# Quantization Robust Federated Learning for Efficient Inference on Heterogeneous Devices

**Kartik Gupta**[†,*]**, Marios Fournarakis**[‡]**, Matthias Reisser**[‡]**, Christos Louizos**[‡]**, Markus Nagel**[‡]
[†] **Australian National University,** [‡] **Qualcomm AI Research**
[†]`kartik.gupta@anu.edu.au`
[‡]`{mfournar,mreisser,clouizos,markusn}@qti.qualcomm.com`

Reviewed on OpenReview: `https://openreview.net/forum?id=lvevdX6bxm`

## Abstract

Federated Learning (FL) is a machine learning paradigm to distributively learn machine learning models from decentralized data that remains on-device. Despite the success of standard Federated optimization methods, such as Federated Averaging (FEDAVG) in FL, the energy demands and hardware induced constraints for on-device learning have not been considered sufficiently in the literature. Specifically, an essential demand for on-device learning is to enable trained models to be quantized to various bit-widths based on the energy needs and heterogeneous hardware designs across the federation. In this work, we introduce multiple variants of federated averaging algorithm that train neural networks robust to quantization. Such networks can be quantized to various bit-widths with only limited reduction in full precision model accuracy. We perform extensive experiments on standard FL benchmarks to evaluate our proposed FEDAVG variants for quantization robustness and provide a convergence analysis for our Quantization-Aware variants in FL. Our results demonstrate that integrating quantization robustness results in FL models that are significantly more robust to different bit-widths during quantized on-device inference.

## 1 Introduction

Federated Learning (FL) is a distributed machine learning paradigm, where a large number of clients, such as consumer smartphones, personal computers or smart home devices learn collaboratively. Clients train on their private local data, which is never shared with other participants in the federation, such as other clients or the server. Despite learning happening on-device, FL results in a single global model at the end of training. The privacy of local client data is an important requirement in modern machine learning and acts as a central motivator for FL.

Several challenges arise in the FL setting. For example, different clients might have different computational constraints based on their hardware design specifications. One practically relevant heterogeneous hardware characteristic is the supported quantization bit-width of the hardware accelerator. A suitably trained model should exhibit no significant performance degradation after quantization to various bit-widths represented in the heterogeneous device landscape. Quantization robustness aims to achieve this objective where a single model is trained with the constraint of robustness to various quantization bit-widths on-the-go at inference time without re-training or finetuning.

Recent works (Shkolnik et al. (2020); Alizadeh et al. (2020); Kim et al. (2020)) introduced novel ways to train quantization robust models in the centralized training setting, but the application of such quantization robustness mechanisms has not been a focus in FL yet. In this work, we address the problem of learning quantization robust models trained using the standard federated learning algorithm, known as Federated

---

*Work done during internship at Qualcomm AI Research. Qualcomm AI Research is an initiative of Qualcomm Technologies, Inc. and/or its subsidiaries.

Averaging (FEDAVG) (McMahan et al. (2017)). To this end, we introduce multiple variants of FEDAVG algorithms to incorporate quantization robustness in FL; this is done either via regularization-based methods for quantization robustness or modified quantization-aware training methods. *Firstly*, we propose the integration of a standard quantization robustness approach known as Kurtosis Regularization (KURE), which involves a regularization term in the local clients' loss function, to enforce uniform distribution on the weights and activations. *Secondly*, we present a quantization robustness approach that involves adding random pseudo-quantization noise during the training procedure. The adopted mechanism is also inspired by the introduction of additive pseudo-quantization noise (Défossez et al. (2021)) for QAT that discards the Straight Through Estimator (STE) approximation (Courbariaux et al. (2015)) for non-differentiable uniform quantization. Straight Through Estimator (STE) approximation is de-facto method in QAT, that allows the backpropagation through a non-differentiable quantization operator by assuming it as an identity function.

Quantization-Aware Training (QAT) methods (Zhou et al. (2016); Krishnamoorthi (2018); Esser et al. (2020); Nagel et al. (2021)) have been successful at training quantized models with ultra-low bit-widths. QAT models perform very well for the target bit-width they have been trained on but can lead to significant degradation for other bit-widths, even for full precision (Kim et al., 2020). To address this limitation of conventional QAT methods, we further introduce Multi-bit Quantization-Aware Training (MQAT), a novel QAT framework that achieves quantized models robust to multiple bit-widths without re-training.

In MQAT for FL, a random bit-width is sampled for each client from the set of considered quantization bit-widths before performing a standard QAT procedure during the client training phase. This small modification enables models trained using the federated regime to be robust to multiple bit-widths during quantized inference. Furthermore, since QAT involves certain heuristics (STE) for computing gradients (due to the non-differentiable rounding operation), we theoretically analyse the convergence behaviour of the global model in the non-convex setting when clients perform local QAT.

Below we summarize the contributions in this paper:

- We introduce multiple quantization robustness methods such as Kurtosis Regularization (KURE), Additive Pseudo-Quantization Noise (APQN) into the federated learning setup to achieve quantization robust models that can be used for efficient inference at multiple bit-widths.

- As the standard form of QAT integration into federated learning fails to generalise across multiple bit-width, we propose Multi-bit Quantization-Aware Training (MQAT), a new variant to achieve quantization robust models in FL.

- We study the theoretical convergence properties for our QAT variants in federated learning and show that the convergence rate for the server-side weights is similar to traditional FL, albeit with a QAT-method-specific error floor.

- We perform extensive experimental evaluations of baselines and our FEDAVG variants on CIFAR-10, CIFAR-100, FEMNIST and TinyImageNet with different network architectures. We empirically demonstrate that our proposed modifications yield models that are robust to quantization at multiple bit-widths without significant reduction on the model's full-precision accuracy.

## 2 Preliminaries

We first provide some brief background on FEDAVG and neural network quantization robustness.

### 2.1 Federated Averaging

Federated learning is a distributed learning paradigm where multiple clients collaboratively learn a shared model. In this machine learning framework, the local client data is not shared with other clients or the server. The problem of federated learning can be formulated as an optimization objective

$$\min_{\mathbf{w}} F(\mathbf{w}) = \mathbb{E}_{i \sim \mathcal{P}}[F_i(\mathbf{w})], \tag{1}$$

where $F_i(\mathbf{w}, \mathcal{D}_i) = \mathbb{E}_{\xi \sim \mathcal{D}_i}[f_i(\mathbf{w}, \xi)]$ is the the local objective function at client $i$, $\mathbf{w} \in \mathbb{R}^D$ represents the parameters for the global model, and $\mathcal{P}$ denotes a distribution over the population of clients $\mathcal{I}$. The local data distribution $\mathcal{D}_i$ often varies between clients, resulting in data heterogeneity.

Federated Averaging (FEDAVG) (McMahan et al. (2017)) is the standard algorithm for federated learning. It operates via a series of *rounds* where each round is divided into a client update phase and server update phase. We denote the total number of rounds as $T$. At the beginning of each round $t$, a subset of clients $\mathcal{S}_t$ is sampled from the pool of clients. The server model is then shared with the sampled clients. During the client update phase, each sampled client runs local SGD for $K$ steps with learning rate $\eta_c$ using their own private data. We denote the $i$-th client's parameters at the $k$-th local step of the $t$-th round by $\mathbf{w}_{t,k}^i$. During the server update phase, the updates of the sampled clients are averaged to build the server-side update $\Delta_t$. The server then applies that update with learning rate $\eta_s$ to receive the next round's parameters $\mathbf{w}_{t+1}$. Reddi et al. (2020) describe a generalization of the server-side update rule to include more advanced adaptive optimizers.

## 2.2 Quantization Robustness

The objective for robust quantization is to learn a single model that can be quantized to different bit-widths without significant degradation in the full precision performance. Given a neural network parameterized by $\mathbf{w}$ that is optimized using a standard loss function $F$, such as the cross entropy, quantization robustness aims at minimizing the following loss:

$$\min_{\mathbf{w}} \quad \sum_{b \in B} F(\mathbf{Q}(\mathbf{w}, b), \mathcal{D}). \tag{2}$$

Here, quantizer $\mathbf{Q}(\cdot)$ with bit-width $b$ and a quantization step size $\Delta_b$, we have that

$$\mathbf{Q}(\mathbf{w}, b) = \Delta_b \cdot \text{clip}\left(\left\lfloor \frac{\mathbf{w}}{\Delta_b} \right\rceil, -2^{b-1}, 2^{b-1} - 1\right), \tag{3}$$

where $\lfloor \cdot \rceil$ denotes the rounding-to-nearest integer operation, and $\text{clip}(\cdot)$ clamps its input such that it lies in the range $[-2^{b-1}, 2^{b-1} - 1]$. The quantization step size can be estimated either post-training or learnt using QAT (Krishnamoorthi (2018); Esser et al. (2020); Nagel et al. (2021)). The above objective intends to learn a neural network that is robust to various bit-widths in the quantization bit set $B$. $B$ could also include the high precision 32-bit floating-point format (FP32). Note that the above formulation explicitly enforces robustness to different bit-widths for weight quantization only. It is straightforward to enforce quantization robustness for activations in a similar manner. Recent works (Alizadeh et al. (2020); Shkolnik et al. (2020)) have explored alternate ways of satisfying the above objective by adding a regularization term in the standard training procedure instead of directly solving the aforementioned optimization problem.

## 3 A Federated Learning Framework with Quantization Robustness

In quantization robust federated learning we aim to solve an optimization problem of the following form:

$$\min_{\mathbf{w}} F(\mathbf{w}) = \mathbb{E}_{i \sim \mathcal{P}}[F_i^*(\mathbf{w})], \tag{4}$$

where $F_i^*(\mathbf{w}, \mathcal{D}_i) = \mathbb{E}_{\xi \sim \mathcal{D}_i} \sum_{b \in B}[f_i(\mathbf{Q}(\mathbf{w}, b), \xi)]$ is a modified local loss to encourage quantization robustness at client $i$ and $B$ is the set of target quantization bit-widths. Instead of directly optimizing this loss, which involves multiple forward-backward passes through the same batch for each bit-width, we incorporate various, more efficient ways for quantization robustness in the FEDAVG framework.

**Regularization Method.** Regularization methods such as Kurtosis regularization (Shkolnik et al. (2020)), which enforce a uniform distribution on the weight tensors, can be incorporated in the FEDAVG framework by modifying the loss function $F_i$ for each client as follows:

$$F_i^*(\mathbf{w}, \mathcal{D}_i) = \mathbb{E}_{\xi \sim \mathcal{D}_i}[f_i(\mathbf{w}, \xi)] + \lambda \cdot L_{\text{KURE}}(\mathbf{w}). \tag{5}$$

The Kurtosis regularization term for an $M$- layered neural network can be expressed as

$$L_{\text{KURE}} = \frac{1}{M} \sum_{i=1}^{M} |\mathcal{K}(\mathbf{w}_i) - \mathcal{K}_\tau|^2, \mathcal{K}(\mathbf{w}) = \mathbb{E}\left[\left(\frac{\mathbf{w} - \mu}{\sigma}\right)^4\right]. \tag{6}$$

Here, $\mu$ and $\sigma$ are the mean and standard deviation of tensor $\mathbf{w}$ and $\mathcal{K}_\tau$ denotes a scalar value that defines the distribution enforced on the tensors. Similar to Shkolnik et al. (2020), we employ $\mathcal{K}_\tau = 1.8$ to enforce uniform distribution. We provide the pseudo-code for FEDAVG with Kurtosis regularization, FEDAVG-KURE, in Algorithm 1.

---

**Algorithm 1** FEDAVG , FEDAVG-KURE

1: **Require** $(\mathbf{w}_0, \eta_c, \eta_s, \lambda)$
2: **for** $t = 0, \ldots, T - 1$ **do**
3:     sample a subset $\mathcal{S}$ of clients
4:     **for all** $i \in \mathcal{S}$ **in parallel do**
5:         $\mathbf{w}_{t,0}^i \leftarrow \mathbf{w}_t$ {broadcast server state to client}
6:         **for** $k = 0, \ldots, K - 1$ **do**
7:             $g_{t,k}^i \leftarrow \nabla f_i(\mathbf{w}_{t,k}^i; \xi_{t,k}^m)$
8:             $f_i^*(\mathbf{w}_{t,k}^i; \xi_{t,k}^m) = f_i(\mathbf{w}_{t,k}^i; \xi_{t,k}^m) + \lambda \cdot L_{\text{KURE}}(\mathbf{w}_{t,k}^i)$
9:             $g_{t,k}^i \leftarrow \nabla f_i^*(\mathbf{w}_{t,k}^i; \xi_{t,k}^m)$
10:           $\mathbf{w}_{t,k+1}^i \leftarrow \mathbf{w}_{t,k}^m - \eta_c \cdot g_{t,k}^i$ {client update}
11:         **end for**
12:     **end for**
13:     $\Delta_t = \frac{1}{|\mathcal{S}|} \sum_{i \in \mathcal{S}} (\mathbf{w}_{t,K}^i - \mathbf{w}_{t,0}^i)$
14:     $\mathbf{w}_{t+1} \leftarrow \mathbf{w}_t + \eta_s \cdot \Delta_t$ {server update}
15: **end for**

---

**Additive Pseudo-Quantization Noise (APQN).** The quantization robustness problem has similarities with adversarial robustness in the sense that both aim to keep predictions unaltered in the presence of some form of bounded additive noise. Adversarially robust models aim to be robust to noised-up input, whereas with quantization robustness the noise is added to either the weight tensor or the intermediate activations. We present a quantization robustness approach that involves adding random pseudo-quantization noise during the training procedure. This is motivated by the recent success of Randomized Smoothing (Cohen et al. (2019)) in the adversarial robustness literature, where a model is learnt with input data samples corrupted with Gaussian noise. An adaptation similar to ours has been presented in the recently introduced Differentiable Quantizer (Défossez et al. (2021)) as a replacement of the commonly used Straight Through Estimator (STE) based quantizer. Their proposed quantizer has only been used to achieve quantized models for a single target bit-width or mixed-precision with fixed computational budget. Since APQN involves additive pseudo quantization noise and the noise does not have any learnable parameters, backpropagation for the rest of the parameters is straightforward.

We propose to learn models that are robust to varying levels of quantization noise and thus can be quantized to different bit-widths. The local loss function in this case can be reformulated as

$$F_i^*(\mathbf{w}, \mathcal{D}_i) = \mathbb{E}_{\xi \sim \mathcal{D}_i}[f_i(\tilde{\mathbf{Q}}(\mathbf{w}, b), \xi)]. \tag{7}$$

Here, $\tilde{\mathbf{Q}}(\cdot)$ is a pseudo-quantizer with bit-width $b$ that adds noise sampled from the uniform distribution $\mathcal{U}[-\Delta_b/2, \Delta_b/2]$, and can be defined as

$$\tilde{\mathbf{Q}}(\mathbf{w}, b) = \mathbf{w} + \mathcal{U}\left[-\frac{\Delta_b}{2}, \frac{\Delta_b}{2}\right]. \tag{8}$$

Provided that $\Delta_b$ is sufficiently large, the sampled pseudo-quantization noise can have support for various target bit-widths, thus encouraging quantization robustness. The pseudo-code for FEDAVG-APQN is provided in Algorithm 2.

**QAT and MQAT.** A standard way of learning a network with low bit-widths is to constrain the parameters and/or activations of the model to fixed quantization levels. This procedure of training a neural network with standard Projected Gradient Descent (PGD) algorithm with quantization function as projection, is often termed Quantization-Aware Training (QAT). To perform "quantization-aware" FL, we can adopt the QAT procedure for the local optimization at each client to learn a global model that can be quantized to a specific bit-width. In this "quantization-aware" FL, the client-level loss function can be reformulated as:

$$F_i^*(\mathbf{w}, \mathcal{D}_i) = \mathbb{E}_{\xi \sim \mathcal{D}_i}[f_i(\mathbf{Q}(\mathbf{w}, b), \xi)]. \tag{9}$$

The quantization step-size $\Delta_b$ can be either learnt as a parameter (Esser et al. (2020)) or be estimated before the start of training and kept fixed thereafter. Note that the backward pass of the network involves a gradient estimate through the non-differentiable rounding operation of $\mathbf{Q}(\cdot)$. To this end, similar to prior QAT literature we use the standard STE approximation (Bengio et al. (2013)), which approximates the gradient of the rounding operator as 1:

$$\frac{\partial \lfloor x \rceil}{\partial x} = 1. \tag{10}$$

By using this STE approximation for QAT, we can train models that can be quantized to specific bit-widths. In order to use the trained model for bit-widths other than $b$, we can analytically estimate the quantization step-size using the ranges for bit-width $b$ as follows:

$$\Delta_a = \frac{2^b - 1}{2^a - 1}\Delta_b. \tag{11}$$

Here, $a$ refers to any target bit-width during inference stage and $b$ refers to bit-width for which the model is trained on. The pseudo-code for FEDAVG-QAT is provided in Algorithm 2.

Although QAT trains models that perform favourably at specific bit-widths, they often suffer from performance degradation when quantizing to other bit-widths (Kim et al. (2020)). For this reason, QAT alone is not suitable for tackling the heterogeneous hardware requirements that one can encounter in a cross-device FL deployment. We propose MQAT to realize quantization-robust FL. MQAT aims to learn models that are robust to a set of bit-widths $B$, by selecting a bitwidth $b \in B$ either randomly during local training or by fixing it based on each client's hardware requirements. The sampled bit-width is then kept same for all the layers.

Similar to QAT, the quantization step-size $\Delta_b$ for different bit-widths can be either learnt or estimated before the start of training and kept fixed thereafter. The quantization step-size for different bit-widths is then shared along the model parameters with all clients. We provide the pseudo-code for FEDAVG-MQAT in Algorithm 2.

**Convergence analysis.** APQN, QAT and MQAT modify the forward pass of the model, either by adding uniform noise in the case of APQN or approximating the gradient of the non-differentiable rounding operation in the case of QAT and MQAT. Therefore, it is important to study how these modifications / heuristics affect the convergence behaviour of the global model in the federated setting. We do not discuss the convergence behaviour of KURE, as KURE uses standard unbiased gradients of a regularized objective and therefore the standard FEDAVG guarantees apply for this objective. What follows now is a convergence analysis in the non-convex setting with the help of the following assumptions:

**Assumption 1** (Lipschitz Gradient). *Each local loss function $F_s$ is $L$-smooth $\forall s \in \mathcal{S}$, i.e., $\|\nabla F_s(\boldsymbol{x}) - \nabla F_s(\boldsymbol{y})\| \leq L\|\boldsymbol{x} - \boldsymbol{y}\|$, $\forall \boldsymbol{x}, \boldsymbol{y} \in \mathbb{R}^D$.*

**Assumption 2** (Bounded variance). *Each $F_s$ has bounded local variance, i.e., $\mathbb{E}[\|\nabla f_s(\mathbf{w}, \epsilon) - \nabla F_s(\mathbf{w})\|^2] \leq \sigma_l^2$, where $f_s$ is a stochastic estimate of the local loss based on a $\mathbf{w} \in \mathbb{R}^D$ and $\epsilon$ is a random mini-batch. Furthermore, the global variance is also bounded, i.e., $\frac{1}{S}\sum_s \|\nabla F_s(\mathbf{w}) - \nabla F(\mathbf{w})\|^2 \leq \sigma_g^2$, $\forall \mathbf{w} \in \mathbb{R}^D$.*

**Assumption 3** (Bounded quantization noise). *Let $\mathbf{w} \in \mathbb{R}^D$, $j$ be any of its dimensions and $Q(\cdot)$ the quantization operation with a step size $\Delta \in \mathbb{R}$. The quantization noise $r_j \in \mathbb{R}$ added to $w_j$, i.e., $r_j = Q(w_j) - w_j$, is bounded by half the step size of $Q(\cdot)$, i.e., $r_j \leq \frac{\Delta}{2}$.*

**Algorithm 2** FEDAVG-APQN , FEDAVG-QAT , and FEDAVG-MQAT

1: **Require** $(\mathbf{w}_0, \eta_c, \eta_s, b, B)$
2: **for** $t = 0, \ldots, T-1$ **do**
3:     sample a subset $\mathcal{S}$ of clients
4:     **for all** $i \in \mathcal{S}$ **in parallel do**
5:         $\mathbf{w}_{t,0}^i \leftarrow \mathbf{w}_t$ {broadcast server state to client}
6:         $b' \leftarrow \mathcal{U}[B]$
7:         **for** $k = 0, \ldots, K-1$ **do**
8:             $g_{t,k}^i \leftarrow \nabla_{\mathbf{w}} f_i(\tilde{\mathbf{Q}}(\mathbf{w}_{t,k}^i, b); \xi_{t,k}^m)$
9:             $g_{t,k}^i \leftarrow \nabla_{\mathbf{w}} f_i(\mathbf{Q}(\mathbf{w}_{t,k}^i, b); \xi_{t,k}^m)$
10:           $g_{t,k}^i \leftarrow \nabla_{\mathbf{w}} f_i(\mathbf{Q}(\mathbf{w}_{t,k}^i, b'); \xi_{t,k}^m)$
11:           $\mathbf{w}_{t,k+1}^i \leftarrow \mathbf{w}_{t,k}^m - \eta_c \cdot g_{t,k}^i$ {client update}
12:         **end for**
13:     **end for**
14:     $\Delta_t = \frac{1}{|\mathcal{S}|} \sum_{i \in \mathcal{S}} (\mathbf{w}_{t,K}^i - \mathbf{w}_{t,0}^i)$
15:     $\mathbf{w}_{t+1} \leftarrow \mathbf{w}_t + \eta_s \cdot \Delta_t$ {server update}
16: **end for**

The first two assumptions are common in the non-convex optimization literature for FL (Reddi et al. (2020)) and the third is automatically satisfied in our quantization schemes, provided the ranges are set up appropriately. Based on these assumptions, we can then make the following statement.

**Theorem 1.** *Let $K$ be the local iterations of each client and $\mathbf{w} \in \mathbb{R}^D$ be the global model parameter vector. Under assumptions 1, 2 and 3, if the client ($\eta_c$) and server ($\eta_s$) learning rates are chosen such that*

$$\eta_c \leq \frac{1}{10LK}, \qquad \eta_c \leq \frac{1}{8LK\eta_s}, \tag{12}$$

*we have that the FEDAVG-{APQN,QAT,MQAT} server updates satisfy*

$$\min_{1 \leq t \leq T} \|\nabla F(\mathbf{w}_t)\|^2 \leq \frac{F(\mathbf{w}_1) - F(\mathbf{w}^*)}{T\eta_s\eta_c A}$$
$$+ \frac{\eta_c}{\eta_s A}(B\sigma_l^2 + \Gamma K\sigma_g^2 + HL^2 DR^2), \tag{13}$$

*where we define*

$$A = \frac{K}{4} - 2L\eta_s\eta_c K^2, \tag{14}$$

$$B = 4\eta_s\eta_c K^2 L^2 + L\eta_s^2\left(2K^2 + \frac{K}{6}\right), \tag{15}$$

$$\Gamma = 24\eta_s\eta_c K^2 L^2 + L\eta_s^2 K, \tag{16}$$

$$H = \frac{4\eta_s}{3\eta_c}K + 6L\eta_s^2 K^2 \tag{17}$$

*and $R = \frac{\Delta_b}{\sqrt{12}}$ for APQN, $R = \frac{\Delta_b}{2}$ for QAT and $R = \sqrt{\mathbb{E}_b\left[\frac{\Delta_b^2}{4}\right]}$ for MQAT.*

The proof is delegated to appendix due to space constraints. It follows Reddi et al. (2020) while handling quantization noise, such as Li et al. (2017). We can see that the convergence rate for the (non-quantized) server side weights is similar to traditional FL (Reddi et al. (2020)), albeit with an additional error floor due to the quantization noise. Some of this quantization error can be reduced by decreasing the learning rates, up to an irreducible factor of $\mathcal{O}(L^2 DR^2)$ that depends on the bit-width. Both the convergence term, *i.e.*, $F(\mathbf{w}_1) - F(\mathbf{w}^*)$, and the the error term scale with the number of local iterations $K$, with the first decaying with $K$ but for the second some of the error terms increase with $K$, due to each step contributing additional quantization noise and drift between the local and the server weights.

## 4 Related Work

**Post-training quantization.** PTQ is fast and efficient way of achieving neural network quantization by using little or no data at inference time. Recent literature in PTQ focused on post-training quantization of LLMs and transformers. Frantar & Alistarh (2022) extend the Optimal Brain Surgeon (OBS) framework to efficiently quantize and prune NNs in a unified setting. Their method is time and space-efficient while achieving high accuracy in vision and language models. Frantar et al. (2023) propose an extension of OBC Frantar & Alistarh (2022) that is optimized for efficient and accurate quantization of generative pretrained models. This one-shot weight quantization method can quantize GPT models with 175 billion parameters in approximately four GPU hours, reducing the bitwidth down to 3 or 4 bits per weight, with negligible accuracy degradation relative to the uncompressed baseline. Liu et al. (2023a) address the issue of quantizing heavy-tailed activations in vision transformers. They discover that for a given quantizer adding a fixed uniform noisy bias to the values being quantized can significantly reduce the quantization error. By adding a noisy bias to each layer they are able to actively alter the activations distribution and make it more quantization-friendly. Yao et al. (2022) proposes a method for efficient and accurate PTQ of large-scale transformers. It comprises of a fine-grained hardware-friendly quantization scheme for both weight and activations employing layer-by-layer knowledge distillation algorithm (LKD) without the access to the original training data. ZeroQuant can achieve 8-bit weight/activation quantization of GPT-3-style models with minimal accuracy impact.

**Quantization-aware training.** QAT is one of the more effective and widely used methods for achieving low-bit weight and activation quantization. It relies on simulating the quantization operation operation during training and requires access to labelled training data. Esser et al. (2020) first introduced the idea of learning the quantization step-size jointly with the weight achieving near floating-point accuracy in ResNets even with 3-bit quantization. Since then, further advances in quantization-aware training (Han et al., 2021; J. Lee, 2021; Gong et al., 2019; Bhalgat et al., 2020; Park & Yoo, 2020) have pushed the envelop and enabled ultra low-bit quantization (2-4 bits) for a wide range of networks and tasks. Recently, Nagel et al. (2022) observe that oscillating latent weights can prevent neural networks from converging to optimal solutions during QAT. They propose freezing oscillating weights or oscillation dampening through regularization and thus improve quantized accuracy in efficient ConvNets. Shin et al. (2023) propose training with pseudo-noise quantization to prevent unstable convergence induced by the straight-through-estimator (STE) in QAT. The NIPQ formulation allows for naturally learning the bitwidth and quantization parameters leading to more accurate and efficient mixed-precision quantized NNs. Liu et al. (2023b) investigate QAT for LLMs and propose a data-free distillation method that leverages generations produced by the pre-trained model, which better preserves the original output distribution and allows quantizing any generative model independent of its training data, similar to PTQ methods. They experiment with LLaMA models of sizes 7B, 13B, and 30B, at quantization levels down to 4 bits.

**Quantization robustness.** A drawback of QAT is that it can make the trained model highly dependent on the chosen bit-width and quantization parameters. To address this, robust quantization aims at training a single set of weights that are robust to a wider range of quantization choices and bit-widths. Alizadeh et al. (2020) model quantization noise as an additive perturbation and show that they can improve quantization robustness by regularizing the $L_1$-norm of gradients. Since this type of gradient regularization is computationally expensive due to the second-order gradient information, a simpler alternative regularization procedure was proposed in follow-up work (Shkolnik et al., 2020). Shkolnik et al. (2020) trains the network with Kurtosis Regularization (KURE) on the weights to improve robustness and use the LAPQ (Nahshan et al., 2020) algorithm to find the optimal quantization parameters post-training. It has been shown in Shkolnik et al. (2020) that the uniform distributions on the weight tensors achieve better quantization robustness, instead of the Gaussian-like distributions attained during the standard training procedure. Recently, Défossez et al. (2021) introduce additive uniform noise to simulate quantization during training for the purpose of QAT; this method can be easily extended for the purpose of quantization robustness as we show in this paper. Kim et al. (2020) introduces a training method that leads to more "quantization-friendly" weights, by scaling the gradient depending on the distance of the weights from the quantization grid.

**Quantization in Federated Learning (FL).** Quantization in the context of FL has been studied mostly in the context of compressing communication, *e.g.*, (Amiri et al., 2020; Reisizadeh et al., 2020; Triastcyn et al., 2021). Equally important to reducing the communication overhead is the reduction of the computational cost of training and inference on-device. Existing work in this direction can roughly be divided into designing more efficient models through *e.g.* sparsification (Caldas et al., 2018b; Jiang et al., 2019; Louizos et al., 2021) and more effective algorithms (Reddi et al., 2020; He et al., 2020) that reduce the number of training rounds. Structurally sparse models can reduce training and inference costs (Horvath et al. (2021) targets heterogeneous computational resources) and may additionally reduce communication (Louizos et al., 2021). Diao et al. (2021) proposes to assign to each client a subset of the global model depending on their resources. A weaker client receives only a subset of hidden layers and as such, it is suited for training as well as inference across a heterogenuous compute landscape. Such an approach is orthogonal to MQAT, similar to how sparsification and quantization are being successfully combined in federated training and centralized settings alike. Quantization, the focus of this work, is orthogonal to these and as of yet understudied in the literature. QuPeL (Ozkara et al., 2021) is very close in scope to our work. A core proposition of their work however is the flexibility of the quantization mechanism to be non-uniform and the set of quantized values that the model weights can occupy to be learnable. Such a method is highly performant in theory but is entirely unsuitable to low-bit accelerators as can be found in today's hardware. Furthermore, QuPeL addresses hardware heterogeneity through personalization, meaning that each client needs to have access a local dataset and perform a finetuning operation in order to select the appropriate centroids for their budget. This is in contrast to our work where the quantization happens "zero-shot", i.e., the client does not need any data (i.e., it could be a new client in the federation, only interested in inference) and can just quantize the server model to a specific, hardware friendly, bit-width. To the best of our knowledge, this work is the first to investigate the robustness of models to different levels of quantization in the FL setting.

## 5 Experiments

We evaluate the quantization robustness of different proposed variants of FEDAVG using standard benchmarks in FL. In this work, we mainly focus on weight quantization robustness of various trained models but we also provide additional experimental comparisons for activation quantization and quantizing both weight and activations. For all the results in the paper, we present the accuracy plots at different quantization bit-widths and we refer the reader for exact numbers to Appendix.

**Experimental Setup.** For the experimental comparisons, we use federated versions of the CIFAR-10, CIFAR-100 (Krizhevsky et al., 2009), TinyImageNet[1] and FEMNIST (Caldas et al., 2018a) datasets. We split the data into 100 (CIFAR-10), 500 (CIFAR-100), 500 (TinyImageNet), 3500 (FEMNIST) clients in a non-i.i.d way following Hsu et al. (2019), where in each round only 10 clients participate for all datasets except TinyImageNet dataset, where 100 clients participate. We train different models for 5000 (CIFAR-10 using ResNet-20), 2000 (CIFAR-10 using LeNet-5), 10000 (CIFAR-100), 4500 (TinyImageNet), and 6000 (FEMNIST) rounds. We use small local batch sizes for all clients in our experiments for different datasets: 64 (CIFAR-10), 20 (CIFAR-100, TinyImageNet, FEMNIST). For all our experiments, we use ADAM optimizer for server training phase and SGD optimizer for client training phase. We use single epoch of local client training for each client participating in a round for all our experiments. For data augmentation, we normalize CIFAR-10 and CIFAR-100 and TinyImageNet to per-channel zero-mean and standard deviation of one. CIFAR-100 further undergoes random cropping to 28 pixels height and width with zero-padding, followed by random horizontal flipping with 50% probability. FEMNIST requires no preprocessing.

In order to simulate a non-i.i.d. data split that we would expect in the federated scenario, we artificially split CIFAR-10, CIFAR-100 and TinyImageNet by their label. For CIFAR-10 and TinyImageNet, the label proportions on each client are computed by sampling from a Dirichlet distribution with $\alpha - 1.0$ (Hsu et al. (2019)). For CIFAR-100 we use the coarse labels provided with the dataset and follow Reddi et al. (2020). For our FEMNIST experiments, the federated split is naturally determined by the writer-id for each client.

---

[1] https://tiny-imagenet.herokuapp.com/

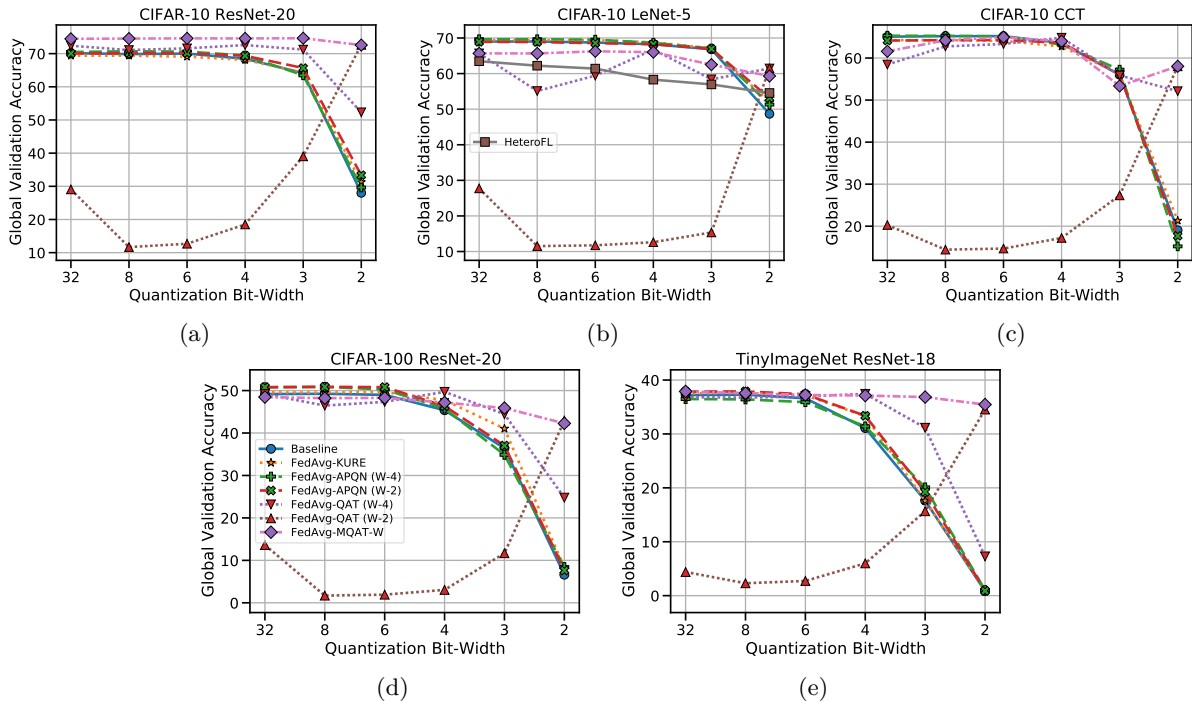

Figure 1: *Global validation accuracy of the proposed* FEDAVG *variants at different bit-widths of weight quantization for models trained on (a) CIFAR-10 using ResNet-20, (b) CIFAR-10 using LeNet-5, (c) CIFAR-100 using ResNet-20, and (d) TinyImageNet using the ResNet-18 architecture. Here, an abbreviation of "W" indicates that we performed weight-quantization only, whereas "W-#" refers to quantization at # bits.*

We use LeNet-5, ResNet-20 and Compact Convolutional Transformers (CCT) (Hassani et al., 2021) architecture for evaluation on CIFAR-10 and LeNet-5 on FEMNIST. For experiments with CIFAR-100 and Tiny-ImageNet, we use ResNet-20 and ResNet-18 network architectures respectively. We replace batch-norm with group-norm in ResNet-20 since batch-norm induces training instabilities in FL (Reddi et al., 2020). We report the accuracy of the quantized server-side model whenever comparing the models at different quantization bit-widths. To compare quantization robustness, we use bit-widths from the set $\{32, 8, 6, 4, 3, 2\}$. We provide the details regarding training hyperparameters in Appendix. For the APQN and KURE methods, we employ post-training quantization methods to set the quantization ranges. Specifically, we find ranges that minimize the mean-squared error (Nagel et al. (2021)). We found that the "Scaler" without "static Batch Normalization" lead to unstable training in LeNet-5. For our HeteroFL (Diao et al., 2021) implementation, we therefore report results without Scaler or Batch Normalization. Batch Normalization negates the scaler's effect, and the absence of Batch Normalization causes the instability. For QAT based methods, range estimation is done at the start of the training and then the same ranges are used at inference time. Our experiments are performed using NVIDIA Tesla V100 GPUs and code is in PyTorch.

## 5.1 Weight Quantization

Firstly, we compare all weight-only quantization-robust FEDAVG variants introduced in this work, *i.e.*, we keep the activations in full precision. We report validation accuracy at the different levels of weight quantization for various datasets and network combinations in Fig. 1.

**CIFAR-10.** Fig. 1a shows the performance of FEDAVG variants on CIFAR-10 with ResNet-20. We see that the classification acurracy for the baseline FEDAVG trained model drops considerably when weights are quantized to low bit-widths (2 and 3). Both, FEDAVG-KURE and FEDAVG-APQN perform similarly to the baseline with only slight improvements at low bit-widths. Our FEDAVG-QAT variants outperforms the other FEDAVG variants. A signfcant drop in validation accuracy is observed as the FEDAVG-QAT

models are quantized to bit-widths other than the target bit-width they have been trained for. This is a known issue with QAT trained models in the context of quantization robustness. Our proposed FEDAVG-MQAT variant directly targets this issue. The FEDAVG-MQAT model outperforms all other variants at different level of quantizations consistently. Furthermore, it improves validation accuracy at full precision as well. Further investigation revealed that for an over-parameterized network such as ResNet-20, the FEDAVG baseline overfits the training set. Our proposed FEDAVG-MQAT implicitly regularizes the FL model and avoids the issue of overfitting in this experimental setup and thus achieves better full precision accuracy. We further investigate overfitting and the implicit regularization phenomenon of FEDAVG-MQAT in Appendix.

Fig. 1b shows performance on CIFAR-10 with the LeNet-5 architecture. Since LeNet-5 is a relatively small network, no overfitting is observed for the FEDAVG baseline. It is important to note that the LeNet-5 (achieves 48-55 % accuracy at 2 bits) is more robust to weight quantization compared to ResNet-20 (achieves 28-35% accuracy at 2 bits) for the baseline FEDAVG trained model on CIFAR-10. Similar to previous comparisons, FEDAVG-KURE and FEDAVG-APQN achieve marginal gains at low bit-widths. The FEDAVG-MQAT produces better validation accuracy (improvement of ≈ 12% at ultra low bit-width of 2 bits but with considerable loss (≈ 3%) in accuracy at full precision. We believe this is because a small network, such as LeNet-5 on CIFAR-10, is inherently hard to compress (quantize) and the gains at low bit-widths come at the cost of considerable degradation in full precision accuracy. Fig. 1c shows performance on CIFAR-10 with the transformer architecture namely CCT (Hassani et al., 2021). Similar to convolutional neural networks, FEDAVG-MQAT consistently outperforms all other other variants at different level of quantizations.

In order to compare our quantization robust mechanism in FL against existing literature, we implemented HeteroFL (Diao et al., 2021). HeteroFL propose to achieve efficient inference by pruning subset of hidden layers based on computer budget. To compare HeteroFL to our proposed methods, we compute the bit operations per second (BOPs) (Van Baalen et al., 2020) for the full-precision activation LeNet-5 at different bit-widths for the weight tensors and chose pruning ratios for HeteroFL that result in the same BOPs. Instead of bits, the HeteroFL curve in Fig. 1b thus presents the evaluation accuracy of the pruned model at different ratios while being close in BOPs to the corresponding quantized LeNet-5 model. We notice that pruning as an alternative approach to quantization generally underperforms quantization for this model and dataset. Since these approaches are orthogonal, we expect future work to explore the combination of quantization and pruning to ensure efficient inference in FL.

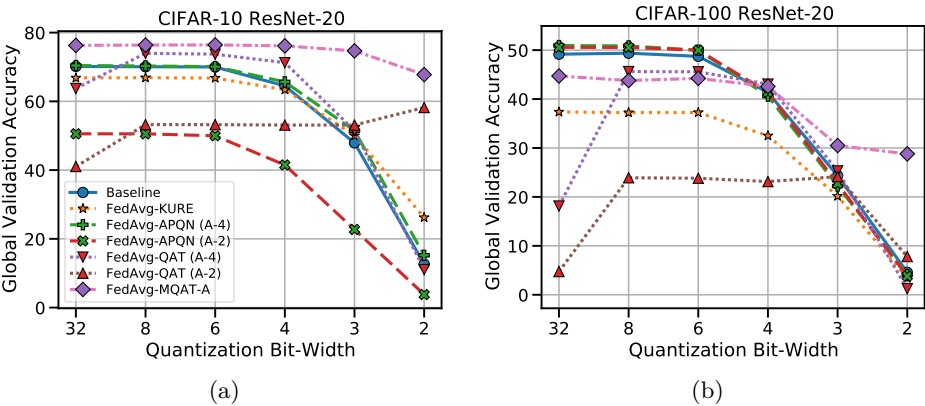

Figure 2: *Global validation accuracy of the proposed FEDAVG variants at different bit-widths of activation quantization for models trained on (a) CIFAR-10 using ResNet-20, and (b) CIFAR-100 using ResNet-20 architecture. Here, an abbreviation of "A" indicates that we performed activation-quantization only, whereas "A-#" refers to quantization at # bits.*

**CIFAR-100.** Fig. 1d shows the performance of our FEDAVG variants with weight-only quantization on CIFAR-100 using the ResNet-20 architecture. Similar to our CIFAR-10 setup, we observe a large drop in the baseline model accuracy at low bits. Our FEDAVG-MQAT variant achieves significant gains, especially at low bit-widths; ≈ 35% at 2-bits and ≈ 10% at 3-bits, compared to the baseline FEDAVG model. It is

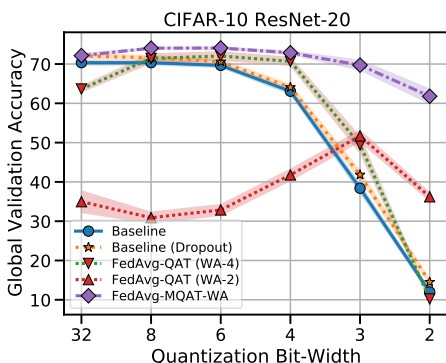

Figure 3: *Global validation accuracy of the proposed FEDAVG variants at different bit-widths of weight and activation quantization for models trained on CIFAR-10 using ResNet-20 architecture. An abbreviation of "WA" indicates that we performed both weight and activation quantization.*

important to note that these improvements on low bit-widths come at the cost of a small loss of accuracy ($\approx 2\%$) at full precision in comparison to the baseline.

**TinyImageNet.** We also performed an experimental evaluation on a more challenging FL setup; classification on TinyImageNet with the ResNet-18 architecture. The results for all of our proposed FEDAVG variants can be seen in Fig. 1e. Despite being the most challenging task with 200 classes and only 500 training samples for each class, our FEDAVG-MQAT variant remains robust to bit-widths as low as 2-bits without any loss in full-precision accuracy. In comparison to the baseline, it improves the 2-bit, 3-bit quantization accuracy by a significant margin ($20-35\%$). We would like to point out that despite being trained for different set of bit-width, our FEDAVG-MQAT variant can achieve accuracy of FEDAVG-QAT for their target bit-widths while preserving the full precision model accuracy.

**FEMNIST.** The FEMNIST dataset has become one of the standard datasets used to evaluate FL algorithms. For the sake of completeness, we performed quantization robustness experiments on FEMNIST using the LeNet-5 architecture. We observe that the baseline FEDAVG trained model is already robust to various quantization levels, even up to 2-bits. Compared to our other tasks, we believe classification on FEMNIST with LeNet-5, is relatively easier and more robust to quantization.

## 5.2 Activation Quantization

To further demonstrate the effectiveness of our FEDAVG variants, we analyze the task of quantizing activations on CIFAR-10 and CIFAR-100 with the ResNet-20 architecture. For FEDAVG-KURE, we impose the regularization term on the activations and, in a similar manner, for FEDAVG-{APQN, QAT and MQAT}, the noise / quantizer is on the activations.

Fig. 2a and Fig. 2b show the validation accuracy at different bit-widths for CIFAR-10 and CIFAR-100 respectively. For both CIFAR-10 and CIFAR-100, we observe considerable decline in the full precision model accuracy after Kurtosis regularization on activations compared to the baseline. It sould be noted that the original work on KURE (Shkolnik et al. (2020)) considered weight-quantization only. FEDAVG-MQAT achieves significant gains at 2-bit ($\approx 55\%$ on CIFAR-10 and $\approx 24\%$ on CIFAR-100) and 3-bit ($\approx 26\%$ on CIFAR-10 and $\approx 6\%$ on CIFAR-100) quantization for both CIFAR-10 and CIFAR-100.

Perhaps surprisingly, we can see that our FEDAVG-MQAT trained model outperforms the respective FEDAVG-QAT models trained on their own respective bitwidth for 2 and 4-bits. For CIFAR-100 dataset, we observe a considerable decrease in model accuracy for FEDAVG-MQAT variant at higher bit-widths (and full precision).

| Bit Config | Federated Averaging (FedAvg) | | |
|---|---|---|---|
| | Baseline | MQAT-W | MQAT*-W |
| W-32 | 70.16 | **74.46** | 70.44 |
| W-8 | 69.86 | **74.54** | 70.88 |
| W-6 | 70.02 | **74.60** | 68.24 |
| W-4 | 68.44 | **74.58** | 70.02 |
| W-3 | 64.14 | **75.64** | 67.58 |
| W-2 | 28.02 | **72.58** | 68.20 |

Table 1: *Global validation accuracy after quantization at various bit-widths for different FEDAVG variants trained on CIFAR-10 dataset using ResNet-20 architecture. Here, * indicates the client-specific bit-width is chosen at the begining of training and then kept fixed throughout.*

### 5.3 Weight and Activation Quantization

The combination of activation and weight quantization promises to fully harness the advantages of specialized hardware accelerators. Fig. 3 illustrates the impact of our quantization robustness variants for joint weight and activation quantization on CIFAR-10 with the ResNet-20 architecture. In this setting, both, weights and activations are quantized before each matrix multiplication. We see that FEDAVG-MQAT outperforms the FEDAVG baseline as well as the bit-specific FEDAVG-QAT across all bit-widths, including full-precision. As noted before, ResNet-20 exhibits overfitting on CIFAR-10. Thus, we include another baseline which introduces dropout (50%) before the final fully-connected layer, which marginally improves performance.

### 5.4 Per-client fixed bit-width.

Performing a forward-pass in low bit-width during Quantization-Aware Training (QAT) does have the additional benefit of reduced computational requirements also during *training*. A good assumption to make is that each client implements efficient hardware acceleration for a specific bit-width that remains constant during its participation in the federated learning process. While in the main experimental setting we considered per-round sampling of the bit-width for MQAT, here we sample a client-specific bit-width at the beginning of training and keep it fixed throughout.

As we can see in Table 1, ex-ante fixed bit-widths lead to some degradation in performance, albeit at the aforementioned benefit of spead-up training.

## 6 Discussion

Real-world deployments of FL, necessarily require catering to a heterogeneous device landscape. Quantization-robust server models, *i.e.*, models that have robust performance on arbitrary target bit-widths, are a step towards effectively navigating such a landscape. In this work, we introduced several variants of the FEDAVG algorithm that encourage quantization-robustness for the server model. Experimentally, we demonstrated that our FEDAVG variants can achieve good performance on several target bit-widths, without significant accuracy degradation for the full precision model. No method clearly outperforms in all settings, although we see MQAT performing well in most situations, especially for lower bit-widths. Theoretically, we showed that quantization-aware local training on the clients provides a convergence rate that is similar to traditional FL, albeit with an extra error floor that depends on the parameters of the quantization procedure and the model characteristics.

In the future, we would focus on end-to-end quantized training which can enable efficiency for both client-server communication as well as on-device training. Further axes of heterogeneity than bit-width should be considered, such as non-uniform quantization, different quantization strategies (symmetric vs. asymmetric) and more subtle differences in inference engines across devices. We believe that the scenario where a client's hardware characteristics are considered fixed throughout training opens up the possibility for advanced client

sub-sampling strategies and necessitates a discussion on the impact of client-specific model performance as a trade-off between efficiency and representation. Device capabilities correlate with the heterogeneous socioeconomic landscape of participating devices and each client's dataset's influence on the learned function will be different.

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

## A Appendix

## Appendix

## B Convergence analysis for quantization aware local training in federated learning

In this appendix we provide a convergence analysis for quantization-aware training in combination with federated learning. We follow the analysis presented in Reddi et al. (2020), while handling the quantization noise, such as Li et al. (2017). This appendix tries to be verbose so as to be easy to follow along. Theorem 3.1 contains the formal claim for this proof. On a high level, we will aim to upper-bound the gradient magnitude of the FL objective (Eq. 1 in the paper), $\nabla_{\mathbf{w}} F(\mathbf{w}_t)$ for different perspectives on $t$. We begin the proof by making explicit assumptions about the objective as well as the nature of quantization. In terms of techniques we rely only on some general inequalities that we detail before the actual proof. We advise the studious reader to keep a second copy of the pdf open to refer back to these inequalities. Furthermore we preface this section by re-iterating notation.

## B.1 Notation and definitions

$\mathbf{w} \in \mathbb{R}^D$    ... is the flattened vector of model parameters.

$\mathbf{w}_{sk}^t \in \mathbb{R}^D$    ... model parameters at the $k$'th (out of $K$) iteration of local updates on client $s$ in communication round $t$.

$\mathcal{S}$    ... is the set of all clients ('shards'), of size $S = |\mathcal{S}|$

$$F(\mathbf{w}) = \frac{1}{S} \sum_s^S F_s(\mathbf{w}), \quad \text{We use } \frac{1}{S} \sum_s F_s(\mathbf{w}) \text{ as shorthand.}$$

$$F_s(\mathbf{w}) = \frac{1}{D_s} \sum_i^{D_s} f_s(\mathbf{w}, \epsilon_i), \quad \text{where } D_s = |\mathcal{D}_s| \text{ is the size of the local dataset } \mathcal{D}_s \text{ at client } s.$$

$f_s(\mathbf{w}, \epsilon_i)$    ... is a loss function evaluated on a random mini-batch $\epsilon_i$ on client $s$ with model parameters $\mathbf{w}$.

$\mathbb{E}_t[F(\mathbf{w}_{t+1})]$    ... is the loss evaluated on parameters $\mathbf{w}$ at round $t + 1$, in expectation over the randomness at round $t$ that influences the transition of parameters from round $t$ to round $t + 1$.

$\mathbb{E}[F(\mathbf{w})]$    ... is the loss evaluated on parameters $\mathbf{w}$, averaged over all possible sources of randomness. Precise definition given in context below.

$\nabla_{\mathbf{w}} F(\mathbf{w})$    ... is the gradient of $F$ with respect to $\mathbf{w}$ evaluated at $\mathbf{w}$. We use the shorthand $\nabla F(\mathbf{w})$, similarly for $\nabla F_s(\mathbf{w})$ and $\nabla f_s(\mathbf{w})$.

$\Delta_b$    ... is the step size of the quantizer / magnitude of the quantization noise for a given bit-width $b$.

## B.2 Assumptions

**Assumption 1** (Lipschitz Gradient). *Each local loss function $F_s$ is $L$-smooth $\forall s \in \mathcal{S}$, i.e., $\|\nabla F_s(\boldsymbol{x}) - \nabla F_s(\boldsymbol{y})\| \leq L\|\boldsymbol{x} - \boldsymbol{y}\|, \forall \boldsymbol{x}, \boldsymbol{y} \in \mathbb{R}^D$.*

**Assumption 2** (Bounded variance). *Each $F_s$ has bounded local variance, i.e., $\mathbb{E}[\|\nabla f_s(\mathbf{w}, \epsilon) - \nabla F_s(\mathbf{w})\|^2] \leq \sigma_l^2$, where $f_s$ is a stochastic estimate of the loss based on an $\mathbf{w} \in \mathbb{R}^D$ and $\epsilon$ is a random mini-batch. Furthermore, the global variance is also bounded, i.e., $\frac{1}{S} \sum_s \|\nabla F_s(\mathbf{w}) - \nabla F(\mathbf{w})\|^2 \leq \sigma_g^2, \forall \mathbf{w} \in \mathbb{R}^D$.*

**Assumption 3** (Bounded quantization noise). *Let $\mathbf{w} \in \mathbb{R}^D$, $j$ be any of its dimensions and $Q(\cdot)$ the quantization operation with a step size $\Delta \in \mathbb{R}$. The quantization noise $r_j \in \mathbb{R}$ added to $w_j$, i.e., $r_j = Q(w_j) - w_j$, is bounded by half the step size of $Q(\cdot)$, i.e., $r_j \leq \frac{\Delta}{2}$.*

## B.3 Auxiliary lemmata / inequalities

In this section we will provide some inequalities and lemmata that will be useful for the proof of our main theorem.

**Lemma 2.** *For any $\gamma > 0$ we have that $\pm 2\alpha\beta \leq \gamma\alpha^2 + \frac{1}{\gamma}\beta^2$*

*Proof.*

$$0 \leq (\sqrt{\gamma}\alpha \pm \frac{1}{\sqrt{\gamma}}\beta)^2 \rightarrow 0 \leq \gamma\alpha^2 + \frac{1}{\gamma}\beta^2 \pm 2\alpha\beta \rightarrow \pm 2\alpha\beta \leq \gamma\alpha^2 + \frac{1}{\gamma}\beta^2.$$

$\square$

**Corollary 3.** *For any $\gamma > 0$ we have that $(\alpha \pm \beta)^2 \leq (1 + \gamma)\alpha^2 + (1 + \frac{1}{\gamma})\beta^2$*

*Proof.*

$$(\alpha \pm \beta)^2 = \alpha^2 + \beta^2 \pm 2\alpha\beta \tag{18}$$

From Lemma 2

$$\leq \alpha^2 + \beta^2 + \gamma\alpha^2 + \frac{1}{\gamma}\beta^2 = (1+\gamma)\alpha^2 + (1+\frac{1}{\gamma})\beta^2 \tag{19}$$

$\square$

**Lemma 4.** *For random variables $\boldsymbol{z}_r, \ldots, \boldsymbol{z}_r$ we have that*

$$\mathbb{E}[\|\boldsymbol{z}_1 + \cdots + \boldsymbol{z}_r\|^2] \leq r\mathbb{E}[\|\boldsymbol{z}_1\|^2 + \cdots + \|\boldsymbol{z}_r\|^2]. \tag{20}$$

*Proof.* The proof follows from expanding the square and applying Lemma 2 to each $2z_i z_j$ term with $\gamma = 1$. $\square$

$$\mathbb{E}_b[\mathbb{E}[\|\mathbf{r}\|^2]] = \sum_d \mathbb{E}_b[\mathbb{E}[r_d^2]] \leq \sum_d \mathbb{E}_b\left[\frac{\Delta_b^2}{4}\right] = \frac{D}{4}\mathbb{E}_b\left[\Delta_b^2\right] \tag{21}$$

**Lemma 5.** *Let $\boldsymbol{r}$ be the quantization noise added to $\mathbf{w}$ satisfying assumption Eq. (3). When performing QAT, MQAT, APQN we have that*

$$\mathbb{E}[\|\boldsymbol{r}\|^2] \leq DR^2, \tag{22}$$

*where $R = \frac{\Delta_b}{2}$ for QAT, $R = \sqrt{\mathbb{E}_b\left[\frac{\Delta_b^2}{4}\right]}$ for MQAT and $R = \frac{\Delta_b}{\sqrt{12}}$ for APQN.*

*Proof.* For QAT we have that

$$\mathbb{E}[\|\boldsymbol{r}\|^2] = \mathbb{E}[\sum_{d=1}^D r_d^2] \leq \mathbb{E}[\sum_{d=1}^D \frac{\Delta_b^2}{4}] = D\left(\frac{\Delta_b}{2}\right)^2, \tag{23}$$

due to the bounded quantization noise assumption. For MQAT where we consider random bitwidths, we have that

$$\mathbb{E}_b[\mathbb{E}[\|\boldsymbol{r}\|^2]] = \sum_d \mathbb{E}_b[\mathbb{E}[r_d^2]] \leq \sum_d \mathbb{E}_b\left[\frac{\Delta_b^2}{4}\right] = D\mathbb{E}_b\left[\frac{\Delta_b^2}{4}\right]. \tag{24}$$

For APQN we have that

$$\mathbb{E}[\|\boldsymbol{r}\|^2] = \mathbb{E}[\sum_{d=1}^D r_d^2] = \sum_{d=1}^D \mathbb{E}[r_d^2] = \sum_{d=1}^D \frac{\Delta_b^2}{12} = D\left(\frac{\Delta_b}{\sqrt{12}}\right)^2, \tag{25}$$

due to $r_d \sim \mathcal{U}\left[-\frac{\Delta_b}{2}, \frac{\Delta_b}{2}\right]$ and $\mathbb{E}[r_d^2] = \text{Var}[r_d] + \mathbb{E}[r_d]^2 = \frac{\Delta_b^2}{12}$. $\square$

**Lemma 6.** *For any local learning rate $\eta_c \leq \frac{1}{10LK}$, we can bound the difference between the local shadow weights and the server weights for any $k \in \{0, \ldots, K-1\}$ and $K \geq 1$ at a given federated training iteration $t$ via*

$$\frac{1}{S}\sum_s \mathbb{E}\|\mathbf{w}_{sk}^t - \mathbf{w}_t\|^2 \leq 4K\eta_c^2(\sigma_l^2 + 6K\sigma_g^2) + 32K^2\eta_c^2L^2DR^2 + 24K^2\eta_c^2\|\nabla F(\mathbf{w}_t)\|^2. \tag{26}$$

*Proof.* We begin by noting that

$$\mathbb{E}\|\mathbf{w}_{sk}^t - \mathbf{w}_t\|^2 = \mathbb{E}\|\underbrace{\mathbf{w}_{s,k-1}^t - \mathbf{w}_t}_{a} - \eta_c(\underbrace{\nabla f_s(\mathbf{w}_{s,k-1}^t + \boldsymbol{r}_{s,k-1}^t) - \nabla F_s(\mathbf{w}_{s,k-1}^t + \boldsymbol{r}_{s,k-1}^t)}_{b})$$

$$+ \underbrace{\nabla F_s(\mathbf{w}_{s,k-1}^t + \boldsymbol{r}_{s,k-1}^t) - \nabla F_s(\mathbf{w}_t)}_{c} + \underbrace{\nabla F_s(\mathbf{w}_t) - \nabla F(\mathbf{w}_t)}_{d} + \underbrace{\nabla F(\mathbf{w}_t)}_{e})\|^2, \qquad (27)$$

where we introduced several shorthand notations for easier manipulation of the terms. By expanding the norm using the multinomial theorem we have that

$$= \mathbb{E}[\|a\|^2] + \eta_c^2\mathbb{E}[\|b\|^2] + \eta_c^2\mathbb{E}[\|c\|^2] + \eta_c^2\mathbb{E}[\|d\|^2] + \eta_c^2\mathbb{E}[\|e\|^2]$$
$$- 2\mathbb{E}[a^T(\eta_c(b + c + d + e))] + 2\eta_c^2\mathbb{E}[b^T(c + d + e)]$$
$$+ 2\eta_c^2\mathbb{E}[c^T(d + e)] + 2\eta_c^2\mathbb{E}[d^T e]. \qquad (28)$$

We can now use the fact that the expectation of $b$ is zero, since $\nabla f_s(\mathbf{w}_{s,k-1}^t + \boldsymbol{r}_{s,k-1}^t)$ is an unbiased estimate of $\nabla F_s(\mathbf{w}_{s,k-1}^t + \boldsymbol{r}_{s,k-1}^t)$. In this way, we can simplify the above to

$$= \mathbb{E}[\|a\|^2] + \eta_c^2\mathbb{E}[\|b\|^2] + \eta_c^2\mathbb{E}[\|c\|^2] + \eta_c^2\mathbb{E}[\|d\|^2] + \eta_c^2\mathbb{E}[\|e\|^2]$$
$$- 2\mathbb{E}[a^T(\eta_c(c + d + e))] + 2\eta_c^2\mathbb{E}[c^T(d + e)] + 2\eta_c^2\mathbb{E}[d^T e]. \qquad (29)$$

We can now use Lemma 2 with a $\gamma = 2K - 1$ in order to "split" the $2\mathbb{E}[a^T(\eta_c(c + d + e))]$ term

$$\leq \mathbb{E}[\|a\|^2] + \eta_c^2\mathbb{E}[\|b\|^2] + \eta_c^2\mathbb{E}[\|c\|^2] + \eta_c^2\mathbb{E}[\|d\|^2] + \eta_c^2\mathbb{E}[\|e\|^2]$$
$$+ \frac{1}{2K - 1}\mathbb{E}[\|a\|^2] + (2K - 1)\eta_c^2\mathbb{E}[\|c + d + e\|^2]$$
$$+ 2\eta_c^2\mathbb{E}[c^T(d + e)] + 2\eta_c^2\mathbb{E}[d^T e]. \qquad (30)$$

Following that, we can see that several terms cancel, due to $\mathbb{E}[\|c + d + e\|^2] = \mathbb{E}[\|c\|^2] + \mathbb{E}[\|d\|^2] + \mathbb{E}[\|e\|^2] + 2\mathbb{E}[c^T(d + e)] + 2\mathbb{E}[d^T e]$

$$= \left(1 + \frac{1}{2K - 1}\right)\mathbb{E}[\|a\|^2] + \eta_c^2\mathbb{E}[\|b\|^2] + 2K\eta_c^2\mathbb{E}[\|c + d + e\|^2]. \qquad (31)$$

Finally, we will apply Lemma 4 in order to split $\mathbb{E}[\|c + d + e\|^2]$ and thus end up with

$$\leq \left(1 + \frac{1}{2K - 1}\right)\mathbb{E}[\|a\|^2] + \eta_c^2\mathbb{E}[\|b\|^2] + 6K\eta_c^2\mathbb{E}[\|c\|^2]$$
$$+ 6K\eta_c^2\mathbb{E}[\|d\|^2] + 6K\eta_c^2\mathbb{E}[\|e\|^2], \qquad (32)$$
$$= \left(1 + \frac{1}{2K - 1}\right)\mathbb{E}[\|\mathbf{w}_{s,k-1}^t - \mathbf{w}_t\|^2]$$
$$+ \eta_c^2\mathbb{E}[\|\nabla f_s(\mathbf{w}_{s,k-1}^t + \boldsymbol{r}_{s,k-1}^t) - \nabla F_s(\mathbf{w}_{s,k-1}^t + \boldsymbol{r}_{s,k-1}^t)\|^2]$$
$$+ 6K\eta_c^2\mathbb{E}[\|\nabla F_s(\mathbf{w}_{s,k-1}^t + \boldsymbol{r}_{s,k-1}^t) - \nabla F_s(\mathbf{w}_t)\|^2]$$
$$+ 6K\eta_c^2\mathbb{E}[\|\nabla F_s(\mathbf{w}_t) - \nabla F(\mathbf{w}_t)\|^2] + 6K\eta_c^2\mathbb{E}[\|\nabla F(\mathbf{w}_t)\|^2], \qquad (33)$$

where we replaced the shorthand notations with their original terms. To proceed, we will make use of assumptions 2 and 1 to arrive at

$$\leq \left(1 + \frac{1}{2K - 1}\right)\mathbb{E}[\|\mathbf{w}_{s,k-1}^t - \mathbf{w}_t\|^2] + \eta_c^2\sigma_l^2$$
$$+ 6K\eta_c^2 L^2\mathbb{E}[\|\mathbf{w}_{s,k-1}^t + \boldsymbol{r}_{s,k-1}^t - \mathbf{w}_t\|^2]$$
$$+ 6K\eta_c^2\mathbb{E}[\|\nabla F_s(\mathbf{w}_t) - \nabla F(\mathbf{w}_t)\|^2] + 6K\eta_c^2\mathbb{E}[\|\nabla F(\mathbf{w}_t)\|^2]. \qquad (34)$$

We can now make use of corollary 3 with a $\gamma = 3$ in order to separate the quantization error $\boldsymbol{r}_{s,k-1}$ from the difference between the local (shadow) weight and the server weight

$$
\begin{aligned}
\leq & \left(1 + \frac{1}{2K-1}\right) \mathbb{E}[\|\mathbf{w}_{s,k-1}^t - \mathbf{w}_t\|^2] + \eta_c^2 \sigma_l^2 \\
& + 24K\eta_c^2 L^2 \mathbb{E}[\|\mathbf{w}_{s,k-1}^t - \mathbf{w}_t\|^2] + 8K\eta_c^2 L^2 \mathbb{E}[\|\boldsymbol{r}_{s,k-1}^t\|^2] \\
& + 6K\eta_c^2 \mathbb{E}[\|\nabla F_s(\mathbf{w}_t) - \nabla F(\mathbf{w}_t)\|^2] + 6K\eta_c^2 \mathbb{E}[\|\nabla F(\mathbf{w}_t)\|^2],
\end{aligned} \tag{35}
$$

and then use Lemma 5 in order to bound the squared norm of $\boldsymbol{r}_{s,k-1}$

$$
\begin{aligned}
\leq & \left(1 + \frac{1}{2K-1}\right) \mathbb{E}[\|\mathbf{w}_{s,k-1}^t - \mathbf{w}_t\|^2] + \eta_c^2 \sigma_l^2 \\
& + 24K\eta_c^2 L^2 \mathbb{E}[\|\mathbf{w}_{s,k-1}^t - \mathbf{w}_t\|^2] + 8K\eta_c^2 L^2 D R^2 \\
& + 6K\eta_c^2 \mathbb{E}[\|\nabla F_s(\mathbf{w}_t) - \nabla F(\mathbf{w}_t)\|^2] + 6K\eta_c^2 \mathbb{E}[\|\nabla F(\mathbf{w}_t)\|^2].
\end{aligned} \tag{36}
$$

To proceed and make further use of our assumptions, we will average the aformentioned inequality over the clients and thus have that

$$
\begin{aligned}
\frac{1}{S}\sum_s \mathbb{E}[\|\mathbf{w}_{sk}^t - \mathbf{w}_t\|^2] \leq & \left(1 + \frac{1}{2K-1} + 24K\eta_c^2 L^2\right) \frac{1}{S}\sum_s \mathbb{E}[\|\mathbf{w}_{s,k-1}^t - \mathbf{w}_t\|^2] + \eta_c^2 \sigma_l^2 \\
& + 8K\eta_c^2 L^2 D Q^2 + 6K\eta_c^2 \frac{1}{S}\sum_s \mathbb{E}[\|\nabla F_s(\mathbf{w}_t) - \nabla F(\mathbf{w}_t)\|^2]
\end{aligned} \tag{37}
$$

$$
+ 6K\eta_c^2 \mathbb{E}[\|\nabla F(\mathbf{w}_t)\|^2], \tag{38}
$$

and then make use of assumption 2 in order to bound the "global" variance

$$
\begin{aligned}
\leq & \left(1 + \frac{1}{2K-1} + 24K\eta_c^2 L^2\right) \frac{1}{S}\sum_s \mathbb{E}[\|\mathbf{w}_{s,k-1}^t - \mathbf{w}_t\|^2] + \eta_c^2 \sigma_l^2 \\
& + 8K\eta_c^2 L^2 D Q^2 + 6K\eta_c^2 \sigma_g^2 + 6K\eta_c^2 \mathbb{E}[\|\nabla F(\mathbf{w}_t)\|^2].
\end{aligned} \tag{39}
$$

Finally, given our assumption that $\eta_c \leq \frac{1}{10LK}$, we have that $(1 + \frac{1}{2K-1} + 24K\eta_c^2 L^2) \leq (1 + \frac{1}{K-1})$ and thus we can simplify the upper bound even further

$$
\begin{aligned}
\frac{1}{S}\sum_s \mathbb{E}[\|\mathbf{w}_{sk}^t - \mathbf{w}_t\|^2] \leq & \left(1 + \frac{1}{K-1}\right) \frac{1}{S}\sum_s \mathbb{E}[\|\mathbf{w}_{s,k-1}^t - \mathbf{w}_t\|^2] + \eta_c^2 (\sigma_l^2 + 6K\sigma_g^2) \\
& + 8K\eta_c^2 L^2 D Q^2 + 6K\eta_c^2 \mathbb{E}[\|\nabla F(\mathbf{w}_t)\|^2].
\end{aligned} \tag{40}
$$

We have now arrived at a point where the average difference between the local shadow weight at iteration $k$ and the server weight is upper bounded by two things; the average difference at iteration $k-1$ along with some constant terms that are independent of the actual weights or iteration. We can thus continue further by sequentially applying the bound at Eq. 40 on each weight difference, up until we end up at the server weight $\mathbf{w}_t$ (since local optimization started from that point) where the difference is zero. Notice that each application of this bound "adds" additional non-negative terms, so we have that the upper bound of $K$ iterations would upper bound the bound on $K-1$ iterations. Therefore, the "worst-case" upper bound is the one where $k = K$. In this case, we can unroll the recursion and have that

$$
\begin{aligned}
\frac{1}{S}\sum_s \mathbb{E}[\|\mathbf{w}_{sk}^t - \mathbf{w}_t\|^2] \leq & \sum_{j=0}^{K-1} \left(1 + \frac{1}{K-1}\right)^j \left(\eta_c^2 (\sigma_l^2 + 6K\sigma_g^2) + 8K\eta_c^2 L^2 D Q^2 \right. \\
& \left. + 6K\eta_c^2 \mathbb{E}[\|\nabla F(\mathbf{w}_t)\|^2]\right).
\end{aligned} \tag{41}
$$

To simplify even further, we can use the fact that $(1 + \frac{1}{K-1})^j$ is monotonic in $j$ and that there are $K$ terms in the sum, thus get

$$
\leq K \left(1 + \frac{1}{K-1}\right)^K \left(\eta_c^2 (\sigma_l^2 + 6K\sigma_g^2) + 8K\eta_c^2 L^2 D Q^2 + 6K\eta_c^2 \mathbb{E}[\|\nabla F(\mathbf{w}_t)\|^2]\right) \tag{42}
$$

and since $(1 + \frac{1}{K-1})^K \leq 4$ for any $K > 1$

$$\leq 4K\eta_c^2(\sigma_l^2 + 6K\sigma_g^2) + 32K^2\eta_c^2 L^2 DQ^2 + 24K^2\eta_c^2 \mathbb{E}[\|\nabla F(\mathbf{w}_t)\|^2], \tag{43}$$

which proves our claim. $\qquad\square$

### B.4   Proof of Theorem 3.1

We begin by noting that the server-side update rule of the model in the case when the clients perform Quantization Aware (QA) SGD with a learning rate of $\eta_c$ and the server does SGD with a learning rate $\eta_s$. The extension of the proof to more involved server-side update rules as in Reddi et al. (2020) is straightforward.

$$\mathbf{w}_{t+1} - \mathbf{w}_t = \eta_s \boldsymbol{G}_t, \qquad \boldsymbol{G}_t = -\frac{\eta_c}{S}\sum_s\sum_k \nabla f_s(\mathbf{w}_{sk}^t + \boldsymbol{r}_{sk}^t), \tag{44}$$

where $s$ indexes the clients and $S$ is the total number of clients. $k$ indexes the local client iteration number, and we assume that there are $K$ local iterations in total. $\mathbf{w}_{sk}^t \in \mathbb{R}^D$ corresponds to a real valued local shadow weight at iteration $t$ and $\boldsymbol{r}_{sk}^t \in \mathbb{R}^D$ corresponds to the quantization noise that is added to it in each iteration of the local optimization (as the weights are rounded / noised before the forward pass).

Using the $L$-smoothness of the global loss function,

$$F(\mathbf{w}_{t+1}) \leq F(\mathbf{w}_t) + \eta_s \nabla F(\mathbf{w}_t)^T \boldsymbol{G}_t + \frac{L}{2}\|\eta_s \boldsymbol{G}_t\|^2 \tag{45}$$

$$= F(\mathbf{w}_t) + \eta_s \nabla F(\mathbf{w}_t)^T \boldsymbol{G}_t + \frac{L\eta_s^2}{2}\|\boldsymbol{G}_t\|^2. \tag{46}$$

We then take an expectation over all randomness at time step $t$,

$$\mathbb{E}_t[F(\mathbf{w}_{t+1})] \leq F(\mathbf{w}_t) + \eta_s \underbrace{\nabla F(\mathbf{w}_t)^T \mathbb{E}_t[\boldsymbol{G}_t]}_{T_{1t}} + \underbrace{\frac{L\eta_s^2}{2}\mathbb{E}_t[\|\boldsymbol{G}_t\|^2]}_{T_{2t}}, \tag{47}$$

and work towards upper bounding the $T_{1t}, T_{2t}$ terms separately.

**Bounding $T_{1t}$**

$$\nabla F(\mathbf{w}_t)^T \mathbb{E}_t[\boldsymbol{G}_t] = \nabla F(\mathbf{w}_t)^T \mathbb{E}_t[\boldsymbol{G}_t - \eta_c K \nabla F(\mathbf{w}_t) + \eta_c K \nabla F(\mathbf{w}_t)] \tag{48}$$

$$= -\eta_c K\|\nabla F(\mathbf{w}_t)\|^2 + \underbrace{\nabla F(\mathbf{w}_t)^T \mathbb{E}_t[\boldsymbol{G}_t + \eta_c K \nabla F(\mathbf{w}_t)]}_{T_{3t}}. \tag{49}$$

We will now work towards upper bounding $T_{3t}$

$$T_{3t} = \nabla F(\mathbf{w}_t)^T \mathbb{E}_t[-\frac{\eta_c}{S}\sum_s\sum_k \nabla f_s(\mathbf{w}_{sk}^t + \boldsymbol{r}_{sk}^t) + \eta_c K \nabla F(\mathbf{w}_t)] \tag{50}$$

$$= \nabla F(\mathbf{w}_t)^T \mathbb{E}_t[-\frac{\eta_c}{S}\sum_s\sum_k \nabla F_s(\mathbf{w}_{sk}^t + \boldsymbol{r}_{sk}^t) + \eta_c K \nabla F(\mathbf{w}_t)] \tag{51}$$

$$= \eta_c \nabla F(\mathbf{w}_t)^T \mathbb{E}_t[-\frac{1}{S}\sum_s\sum_k \nabla F_s(\mathbf{w}_{sk}^t + \boldsymbol{r}_{sk}^t) + \frac{1}{S}\sum_s\sum_k \nabla F_s(\mathbf{w}_t)]. \tag{52}$$

Now by using Lemma 2 with $\gamma = K$ we have that

$$T_{3t} \leq \frac{\eta_c K}{2}\|\nabla F(\mathbf{w}_t)\|^2 + \frac{\eta_c}{2K}\mathbb{E}_t[\|\frac{1}{S}\sum_s\sum_k \nabla F_s(\mathbf{w}_{sk}^t + \boldsymbol{r}_{sk}^t) - \frac{1}{S}\sum_s\sum_k \nabla F_s(\mathbf{w}_t)\|^2] \tag{53}$$

$$= \frac{\eta_c K}{2}\|\nabla F(\mathbf{w}_t)\|^2 + \frac{\eta_c}{2KS^2}\mathbb{E}_t[\|\sum_s\sum_k (\nabla F_s(\mathbf{w}_{sk}^t + \boldsymbol{r}_{sk}^t) - \nabla F_s(\mathbf{w}_t))\|^2]. \tag{54}$$

By then using Lemma 4 to push the squared norm inside the sum

$$\leq \frac{\eta_c K}{2}\|\nabla F(\mathbf{w}_t)\|^2 + \frac{\eta_c SK}{2KS^2}\mathbb{E}_t[\sum_s\sum_k\|\nabla F_s(\mathbf{w}_{sk}^t + \boldsymbol{r}_{sk}^t) - \nabla F_s(\mathbf{w}_t))\|^2] \tag{55}$$

and from the Lipschitz gradient assumption

$$\leq \frac{\eta_c K}{2}\|\nabla F(\mathbf{w}_t)\|^2 + \frac{\eta_c}{2S}\mathbb{E}_t[\sum_s\sum_k\|L(\mathbf{w}_{sk}^t + \boldsymbol{r}_{sk}^t - \mathbf{w}_t)\|^2] \tag{56}$$

$$= \frac{\eta_c K}{2}\|\nabla F(\mathbf{w}_t)\|^2 + \frac{\eta_c L^2}{2}\sum_k\frac{1}{S}\sum_s\mathbb{E}_t[\|\mathbf{w}_{sk}^t + \boldsymbol{r}_{sk}^t - \mathbf{w}_t\|^2]. \tag{57}$$

We will now once more use Lemma 4 to separate the shadow weight difference from the quantization noise

$$\leq \frac{\eta_c K}{2}\|\nabla F(\mathbf{w}_t)\|^2 + \eta_c L^2\sum_k\frac{1}{S}\sum_s\mathbb{E}_t[\|\mathbf{w}_{sk}^t - \mathbf{w}_t\|^2] + \eta_c L^2\sum_k\frac{1}{S}\sum_s E_t[\|\boldsymbol{r}_{sk}^t\|^2] \tag{58}$$

and from Lemma 5

$$\leq \frac{\eta_c K}{2}\|\nabla F(\mathbf{w}_t)\|^2 + \eta_c L^2\sum_k\frac{1}{S}\sum_s\mathbb{E}_t[\|\mathbf{w}_{sk}^t - \mathbf{w}_t\|^2] + \eta_c KL^2 DR^2. \tag{59}$$

We see that in order to proceed, we need to upper bound the difference between the local shadow weight at any iteration $k$ and the server weight. This is were we will use our Lemma 6 in order to proceed

$$\leq \frac{\eta_c K}{2}\|\nabla F(\mathbf{w}_t)\|^2 + K\eta_c L^2(4K\eta_c^2(\sigma_l^2 + 6K\sigma_g^2)) + K\eta_c L^2(32K^2\eta_c^2 L^2 DQ^2)$$
$$+ K\eta_c L^2(24K^2\eta_c^2\|\nabla F(\mathbf{w}_t)\|^2) + \eta_c KL^2 DR^2 \tag{60}$$

$$= \left(\frac{\eta_c K}{2} + 24K^3\eta_c^3 L^2\right)\|\nabla F(\mathbf{w}_t)\|^2 + 4K^2\eta_c^3 L^2(\sigma_l^2 + 6K\sigma_g^2) + (\eta_c K + 32\eta_c^3 K^3 L^2)L^2 DR^2. \tag{61}$$

Finally, we can make use of our assumption $\eta_c \leq \frac{1}{10LK}$ which leads to $24\eta_c^3 L^2 K^3 \leq \frac{1}{4}\eta_c K$ and $32\eta_c^3 L^2 K^3 \leq \frac{1}{3}\eta_c K$. In this way, we can arrive at our final bound for $T_{3t}$

$$T_{3t} \leq \frac{3\eta_c K}{4}\|\nabla F(\mathbf{w}_t)\|^2 + \frac{4}{3}\eta_c KL^2 DR^2 + 4K^2\eta_c^3 L^2(\sigma_l^2 + 6K\sigma_g^2). \tag{62}$$

Now we can apply this bound to $T_{1t}$ in order to get

$$T_{1t} \leq -\eta_c K\|\nabla F(\mathbf{w}_t)\|^2 + \frac{3\eta_c K}{4}\|\nabla F(\mathbf{w}_t)\|^2 + \frac{4}{3}\eta_c KL^2 DR^2 + 4K^2\eta_c^3 L^2(\sigma_l^2 + 6K\sigma_g^2) \tag{63}$$

$$= -\frac{\eta_c K}{4}\|\nabla F(\mathbf{w}_t)\|^2 + \frac{4}{3}\eta_c KL^2 DR^2 + 4K^2\eta_c^3 L^2(\sigma_l^2 + 6K\sigma_g^2) \tag{64}$$

Which is the final upper bound on $T_{1t}$.

**Bounding $T_{2t}$**  We begin by noting that

$$\frac{L\eta_s^2}{2}\mathbb{E}_t[\|\boldsymbol{G}_t\|^2] = \frac{L\eta_s^2}{2}\mathbb{E}_t[\|\boldsymbol{G}_t - \eta_c K\nabla F(\mathbf{w}_t) + \eta_c K\nabla F(\mathbf{w}_t)\|^2], \tag{65}$$

and then we can split the squared norm via Corollary 3 with $\gamma = 1$

$$\leq L\eta_s^2(\underbrace{\mathbb{E}_t[\|\boldsymbol{G}_t + \eta_c K\nabla F(\mathbf{w}_t)\|^2]}_{T_{4t}} + \eta_c^2 K^2\|\nabla F(\mathbf{w}_t)\|^2). \tag{66}$$

To continue, we will move towards upper bounding $T_{4t}$ by expanding the terms inside the squared norm

$$T_{4t} = \mathbb{E}_t[\|-\frac{\eta_c}{S}\sum_s\sum_k\nabla f_s(\mathbf{w}_{sk}^t + \boldsymbol{r}_{sk}^t) + \frac{\eta_c}{S}\sum_s\sum_k\nabla F_s(\mathbf{w}_t)\|^2] \tag{67}$$

$$= \frac{\eta_c^2}{S^2}\mathbb{E}_t[\|\sum_s\sum_k\left(\nabla f_s(\mathbf{w}_{sk}^t + \boldsymbol{r}_{sk}^t) - \nabla F_s(\mathbf{w}_{sk}^t + \boldsymbol{r}_{sk}^t) + \nabla F_s(\mathbf{w}_{sk}^t + \boldsymbol{r}_{sk}^t) - \nabla F_s(\mathbf{w}_t)\right)\|^2]. \tag{68}$$

We will then apply Lemma 4 in order to move the squared norm inside the sums

$$\leq \frac{\eta_c^2 SK}{S^2} \sum_s \sum_k \mathbb{E}_t[\|\nabla f_s(\mathbf{w}_{sk}^t + \boldsymbol{r}_{sk}^t) - \nabla F_s(\mathbf{w}_{sk}^t + \boldsymbol{r}_{sk}^t) + \nabla F_s(\mathbf{w}_{sk}^t + \boldsymbol{r}_{sk}^t) - \nabla F_s(\mathbf{w}_t)\|^2] \quad (69)$$

and will apply Corollary 3 with $\gamma = 1$ in order to split the norm

$$\leq \frac{2\eta_c^2 K}{S} \sum_s \sum_k \left( \mathbb{E}_t[\|\nabla f_s(\mathbf{w}_{sk}^t + \boldsymbol{r}_{sk}^t) - \nabla F_s(\mathbf{w}_{sk}^t + \boldsymbol{r}_{sk}^t)\|^2] + \mathbb{E}_t[\|\nabla F_s(\mathbf{w}_{sk}^t + \boldsymbol{r}_{sk}^t) \right.$$
$$\left. - \nabla F_s(\mathbf{w}_t)\|^2] \right). \quad (70)$$

To proceed, we will make use of our assumptions 1, 2 in order to get

$$\leq 2\eta_c^2 K^2 \sigma_l^2 + 2\eta_c^2 K \sum_k \frac{1}{S} \sum_s \mathbb{E}_t[\|L(\mathbf{w}_{sk}^t + \boldsymbol{r}_{sk}^t - \mathbf{w}_t)\|^2] \quad (71)$$

and we will apply Corollary 3 with $\gamma = 1$ one more time in order to split the norm of the weight difference and the quantization error

$$\leq 2\eta_c^2 K^2 \sigma_l^2 + 4\eta_c^2 KL^2 \sum_k \frac{1}{S} \sum_s \mathbb{E}_t[\|\mathbf{w}_{sk}^t - \mathbf{w}_t\|^2] + 4\eta_c^2 KL^2 \sum_k \frac{1}{S} \sum_s \mathbb{E}_t[\|\boldsymbol{r}_{sk}^t\|^2] \quad (72)$$

so that we can apply Lemma 5 in order to bound the latter

$$\leq 2\eta_c^2 K^2 \sigma_l^2 + 4\eta_c^2 KL^2 \sum_k \frac{1}{S} \sum_s \mathbb{E}_t[\|\mathbf{w}_{sk}^t - \mathbf{w}_t\|^2] + 4\eta_c^2 K^2 L^2 DR^2. \quad (73)$$

By observing the above, we see that we again end up with the average difference between the shadow weights at each iteration $k$ and the server weight. As a result, we can apply Lemma 6 in order to proceed further

$$\leq 2\eta_c^2 K^2 \sigma_l^2 + 4\eta_c^2 K^2 L^2 DQ^2$$
$$+ 4\eta_c^2 K^2 L^2 (4K\eta_c^2(\sigma_l^2 + 6K\sigma_g^2) + 32K^2\eta_c^2 L^2 DR^2 + 24K^2\eta_c^2 \mathbb{E}[\|\nabla F(\mathbf{w}_t)\|^2]) \quad (74)$$
$$= 2\eta_c^2 K^2 \sigma_l^2 + 4\eta_c^2 K^2 L^2 DQ^2 + 16K^3\eta_c^4 L^2(\sigma_l^2 + 6K\sigma_g^2) + 128\eta_c^4 L^4 K^4 DR^2$$
$$+ 96\eta_c^4 K^4 L^2 \|\nabla F(\mathbf{w}_t)\|^2. \quad (75)$$

In order to simplify the aforementioned inequality we will make a use of our assumption on $\eta_c$, namely that $\eta_c \leq \frac{1}{10LK}$. In this way, we will have that $16K^3\eta_c^4 L^2 \leq \frac{1}{6}K\eta_c^2$ along with $128\eta_c^4 K^4 L^2 \leq \frac{3}{2}\eta_c^2 K^2$. Taking these into account, we have that

$$\leq (2\eta_c^2 K^2 + \frac{1}{6}K\eta_c^2)\sigma_l^2 + K^2\eta_c^2\sigma_g^2 + (4\eta_c^2 K^2 + \frac{3}{2}\eta_c^2 K^2)L^2 DR^2 + 96\eta_c^4 K^4 L^2 \|\nabla F(\mathbf{w}_t)\|^2 \quad (76)$$

and due to $4 + \frac{3}{2} < 6$

$$\leq (2\eta_c^2 K^2 + \frac{1}{6}K\eta_c^2)\sigma_l^2 + K^2\eta_c^2\sigma_g^2 + 6\eta_c^2 K^2 L^2 DR^2 + 96\eta_c^4 K^4 L^2 \|\nabla F(\mathbf{w}_t)\|^2, \quad (77)$$

which constitutes our final upper bound on $T_{4t}$. With this bound at hand, we can move back to bounding $T_{2t}$ and thus get

$$T_{2t} \leq L\eta_s^2 ((2\eta_c^2 K^2 + \frac{1}{6}K\eta_c^2)\sigma_l^2 + K^2\eta_c^2\sigma_g^2 + 6\eta_c^2 K^2 L^2 DR^2 + 96\eta_c^4 K^4 L^2 \|\nabla F(\mathbf{w}_t)\|^2$$
$$+ \eta_c^2 K^2 \|\nabla F(\mathbf{w}_t)\|^2) \quad (78)$$
$$= L\eta_s^2 ((2\eta_c^2 K^2 + \frac{1}{6}K\eta_c^2)\sigma_l^2 + K^2\eta_c^2\sigma_g^2 + 6\eta_c^2 K^2 L^2 DR^2)$$
$$+ L\eta_s^2 (96\eta_c^4 K^4 L^2 + \eta_c^2 K^2) \|\nabla F(\mathbf{w}_t)\|^2. \quad (79)$$

Having bounded $T_{1t}$ and $T_{2t}$, we can now apply these bounds to the inequality at Eq. 47 and thus get

$$\mathbb{E}_t[F(\mathbf{w}_{t+1})] \leq F(\mathbf{w}_t) + \eta_s \underbrace{\nabla F(\mathbf{w}_t)^T \mathbb{E}_t[\boldsymbol{G}_t]}_{T_{1t}} + \underbrace{\frac{L\eta_s^2}{2} \mathbb{E}_t[\|\boldsymbol{G}_t\|^2]}_{T_{2t}} \tag{80}$$

$$\leq F(\mathbf{w}_t) - \frac{\eta_s \eta_c K}{4} \|\nabla F(\mathbf{w}_t)\|^2 + \frac{4}{3}\eta_s \eta_c K L^2 D R^2 + 4\eta_s K^2 \eta_c^3 L^2 (\sigma_l^2 + 6K\sigma_g^2)$$
$$+ L\eta_s^2 ((2\eta_c^2 K^2 + \frac{1}{6}K\eta_c^2)\sigma_l^2 + K^2 \eta_c^2 \sigma_g^2 + 6\eta_c^2 K^2 L^2 D R^2)$$
$$+ L\eta_s^2 (96\eta_c^4 K^4 L^2 + \eta_c^2 K^2)\|\nabla F(\mathbf{w}_t)\|^2. \tag{81}$$

To simplify the aforementioned bound, we can once again make use of our condition $\eta_c \leq \frac{1}{10LK}$ which leads to $96\eta_c^4 K^4 L^2 \leq \eta_c^2 K^2$. In this way, we get that

$$\mathbb{E}_t[F(\mathbf{w}_{t+1})] \leq F(\mathbf{w}_t) - \eta_s(\frac{\eta_c K}{4} - 2L\eta_s \eta_c^2 K^2)\|\nabla F(\mathbf{w}_t)\|^2$$
$$+ (4\eta_s K^2 L^2 \eta_c^3 + L\eta_s^2(2\eta_c^2 K^2 + \frac{1}{6}K\eta_c^2))\sigma_l^2$$
$$+ (24\eta_s K^2 L^2 \eta_c^3 + L\eta_s^2 \eta_c^2 K)K\sigma_g^2 + (\frac{4}{3}\eta_s \eta_c K + 6L\eta_s^2 \eta_c^2 K^2)L^2 D R^2 \tag{82}$$

$$= F(\mathbf{w}_t) - \eta_s \eta_c \underbrace{(\frac{K}{4} - 2L\eta_s \eta_c K^2)}_{A} \|\nabla F(\mathbf{w}_t)\|^2$$
$$+ \eta_c^2 \underbrace{(4\eta_s K^2 L^2 \eta_c + L\eta_s^2(2K^2 + \frac{K}{6}))}_{B} \sigma_l^2$$
$$+ \eta_c^2 \underbrace{(24\eta_s K^2 L^2 \eta_c + L\eta_s^2 K)}_{\Gamma} K\sigma_g^2 + \eta_c^2 \underbrace{(\frac{4\eta_s}{3\eta_c}K + 6L\eta_s^2 K^2)}_{H} L^2 D R^2, \tag{83}$$

where we introduced several shorthand notations for easier manipulation of the inequalities. We can thus now re-arrange the inequality to

$$\mathbb{E}_t[F(\mathbf{w}_{t+1})] - F(\mathbf{w}_t) \leq -\eta_s \eta_c A\|\nabla F(x_t)^2\| + \eta_c^2(B\sigma_l^2 + \Gamma K\sigma_g^2) + \eta_c^2 H L^2 D R^2. \tag{84}$$

In order to consider the entire training trajectory, we will use a telescoping sum, *i.e.*, we will sum this inequality over all rounds and take the expectation at each time-step

$$\sum_{t=1}^{T}(\mathbb{E}_t[F(\mathbf{w}_{t+1})] - \mathbb{E}_{t-1}[F(\mathbf{w}_t)]) \leq -\eta_s \eta_c A \sum_{t=1}^{T} \|\nabla F(\mathbf{w}_t)\|^2 + T\eta_c^2(B\sigma_l^2 + \Gamma K\sigma_g^2)$$
$$+ T\eta_c^2 H L^2 D R^2. \tag{85}$$

In doing that, most of the terms on the left-hand-side will cancel across subsequent time-steps and thus we will end up with

$$\mathbb{E}_T[F(\mathbf{w}_{T+1})] - F(\mathbf{w}_1) \leq -\eta_s \eta_c A \sum_{t=1}^{T} \|\nabla F(\mathbf{w}_t)\|^2 + T\eta_c^2(B\sigma_l^2 + \Gamma K\sigma_g^2) + T\eta_c^2 H L^2 D R^2. \tag{86}$$

We can now re-arrange the terms to get

$$\eta_s \eta_c A \sum_{t=1}^{T} \|\nabla F(\mathbf{w}_t)\|^2 \leq F(\mathbf{w}_1) - \mathbb{E}_T[F(\mathbf{w}_{T+1})] + T\eta_c^2(B\sigma_l^2 + \Gamma K\sigma_g^2) + T\eta_c^2 H L^2 D R^2. \tag{87}$$

| Dataset | Network | $\eta_s$ | $\eta_c$ | $\epsilon_s$ | **Rounds** $(T)$ | **Batch Size** |
|---|---|---|---|---|---|---|
| CIFAR-10 | LeNet-5 | $1e-3$ | $5e-2$ | $1e-8$ | 2000 | 64 |
| CIFAR-10 | ResNet-20 | $1e-3$ | $5e-2$ | $1e-7$ | 5000 | 64 |
| CIFAR-100 | ResNet-20 | $1e-3$ | $5e-2$ | $1e-7$ | 10000 | 20 |
| FEMNIST | LeNet-5 | $1e-3$ | $5e-2$ | $1e-8$ | 6000 | 20 |
| TinyImageNet | ResNet-18 | $1e-2$ | $1e-2$ | $1e-3$ | 4500 | 20 |

Table 2: *Hyperparameters used for the experimental evaluations in the paper. Here $\eta_s$, $\eta_c$ denote the server and client learning rate and $\epsilon_s$ refers to the correction term in value in ADAM optimizer of server.*

and by considering $\mathbf{w}^*$ to be the parameters of lowest loss, $\mathbb{E}_T[F(\mathbf{w}_{T+1})] \geq F(\mathbf{w}^*)$, we have that

$$\eta_s \eta_c A \sum_{t=1}^{T} \|\nabla F(\mathbf{w}_t)\|^2 \leq F(\mathbf{w}_1) - F(\mathbf{w}^*) + T\eta_c^2(B\sigma_l^2 + \Gamma K \sigma_g^2) + T\eta_c^2 H L^2 D R^2. \tag{88}$$

In order to proceed, we have to impose a condition on $A$, namely that it has to be positive (otherwise, dividing by $A$ reverses the inequality). For this to happen we need that

$$\frac{K}{4} \geq 2L\eta_s\eta_c K^2 \rightarrow \frac{1}{4} \geq 2L\eta_s\eta_c K \rightarrow \eta_c \leq \frac{1}{8LK\eta_s}. \tag{89}$$

Assuming that this condition is satisfied, we have that

$$\frac{1}{T}\sum_{t=1}^{T}\|\nabla F(\mathbf{w}_t)\|^2 \leq \frac{F(\mathbf{w}_1) - F(\mathbf{w}^*)}{T\eta_s\eta_c A} + \frac{\eta_c}{\eta_s A}(B\sigma_l^2 + \Gamma K\sigma_g^2 + HL^2 DR^2). \tag{90}$$

$$\tag{91}$$

Finally, in order to obtain the result of Theorem 3.1, we make use of the fact that $\min_{1 \leq t \leq T}\|\nabla F(\mathbf{w}_t)\|^2 \leq \frac{1}{T}\sum_{t=1}^{T}\|\nabla F(\mathbf{w}_t)\|^2$ and thus arrive at

$$\min_{1 \leq t \leq T}\|\nabla F(\mathbf{w}_t)\|^2 \leq \frac{F(\mathbf{w}_1) - F(\mathbf{w}^*)}{T\eta_s\eta_c A} + \frac{\eta_c}{\eta_s A}(B\sigma_l^2 + \Gamma K\sigma_g^2 + HL^2 DR^2), \tag{92}$$

which completes the proof.

## C   Final Hyperparameters

We provide the final hyperparameters used for all the experiments in Table 2. Since the grid-search for FL is expensive, we tune the hyperparameters such as client learning rate ($\eta_c$), server learning rate ($\eta_s$) and epsilon term ($\epsilon_s$) used in ADAM optimizer only for the baselines and then use the same set of hyperparameters for all our proposed variants. For FEDAVG-KURE, we tuned $\lambda$ from the grid $[1e+1, 1e+0, 1e-1, 1e-2, 1e-3, 1e-4]$, found $\lambda = 1e-1$ to be optimal and use that for all our experiments. We also provide the details about bit-set used to perform FEDAVG-MQAT variants in Table 3.

## D   Additional Results

**Implicit Regularization of MQAT.**   As discussed in the main paper, ResNet-20 on CIFAR-10 exhibits overfitting, which we further investigate here. It was observed in the main text that FEDAVG-MQAT has an implicit regularisation effect and thus achieves better validation accuracy for the full-precision model by avoiding overfitting despite that not being the main objective. We show experimental comparisons of our proposed MQAT to other standard methods of regularization with different strength of weight decay (measured by regularisation term $\lambda_{\text{WD}}$) and drop-out before the last full-connected layer of the network in

| Dataset | Network | Federated Averaging (FedAvg) | | |
|---|---|---|---|---|
| | | **MQAT-W** | **MQAT-A** | **MQAT-WA** |
| CIFAR-10 | LeNet-5 | [4,6,8] | - | - |
| CIFAR-10 | ResNet-20 | [2,3,4,6,8,32] | [2,3,4,6,8,32] | [2,3,4,6,8,32] |
| CIFAR-100 | ResNet-20 | [2,3,4,6,8,32] | [2,3,4,6,8,32] | - |
| TinyImageNet | ResNet-18 | [2,3,4,6,8] | - | - |

Table 3: *Quantization Bit-Set used to perform FEDAVG-MQAT on different experimental setup in the paper.*

Table 4. Despite not being its primary objective, our FEDAVG-MQAT variant achieves considerable gains in full-precision accuracy in comparison to standard approaches used to avoid overfitting. Fig. 4 shows how the validation loss of the baseline is increasing after $2.5k$ rounds and the corresponding validation accuracy on the right. For MQAT, we observe no overfitting according to the validation loss.

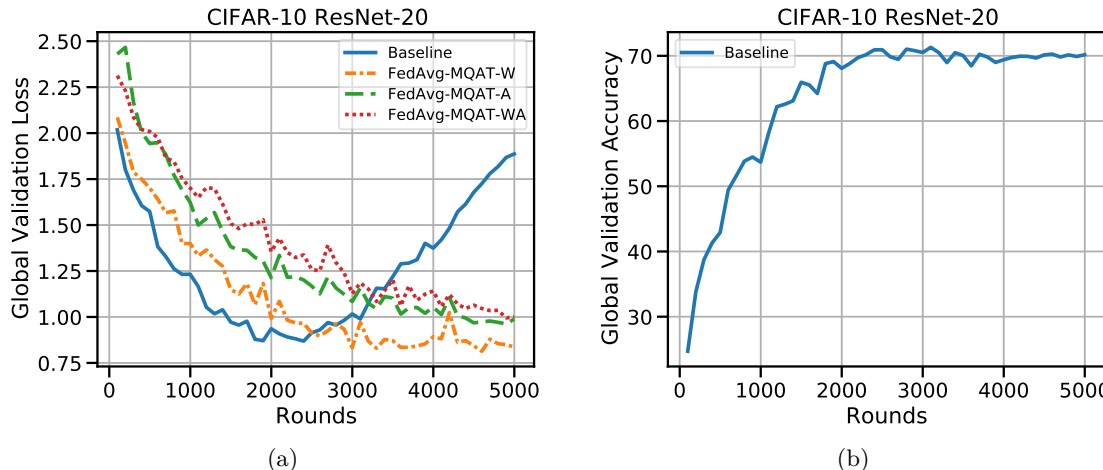

Figure 4: *Global validation (a) loss and (b) accuracy curves for Baseline and different FEDAVG-MQAT variants trained on CIFAR-100 using ResNet-20 architecture. For FEDAVG-MQAT variants the validation loss refers to the loss after quantization at lowest bit-width in the bit-set B. While Baseline model clearly suffers from the overfitting issue, our FEDAVG-MQAT variants clearly manage to avoid it. Further, the validation accuracy curves on the baselines reveals that overfitting cannot be prevented even by early stopping.*

**Detailed results from the main text** In this section, we provide the exact values used for plotting various figures in the main text in Table 5-11.

| FedAvg | Accuracy |
|---|---|
| Baseline | 70.16 |
| Baseline (Dropout) | 72.02 |
| $\lambda_{WD} = 1e-2$ | 44.94 |
| $\lambda_{WD} = 1e-3$ | 71.62 |
| $\lambda_{WD} = 1e-4$ | 70.76 |
| $\lambda_{WD} = 1e-5$ | 70.06 |
| $\lambda_{WD} = 1e-6$ | 70.10 |
| MQAT-W | 74.46 |
| MQAT-A | 76.28 |
| MQAT-WA | 74.90 |

Table 4: *Global validation accuracy for full-precision models learnt in federation using FEDAVG variants trained on CIFAR-10 dataset with ResNet-20 architecture. Here W indicates the client-specific bit-width is chosen at the begining of training and then kept fixed throughout. Here, an abbreviation of "W", "A" and "WA" indicate weight quantization, activation quantization and both weight-activation.*

| Bit Config | Federated Averaging (FedAvg) | | | | | | |
|---|---|---|---|---|---|---|---|
| | Baseline | KURE | APQN (W-4) | APQN (W-2) | QAT (W-4) | QAT (W-2) | MQAT-W |
| W-32 | 70.16 | 69.38 | 70.5 | 70.04 | 72.38 | 29.06 | 74.46 |
| W-8 | 69.86 | 69.38 | 70.72 | 70.19 | 71.08 | 11.68 | 74.54 |
| W-6 | 70.02 | 69.08 | 70.62 | 70.16 | 71.6 | 12.68 | 74.60 |
| W-4 | 68.44 | 68.12 | 69.44 | 69.36 | 72.58 | 18.5 | 74.58 |
| W-3 | 64.14 | 64.28 | 63.46 | 65.66 | 71.22 | 39.02 | 74.64 |
| W-2 | 28.02 | 31.28 | 29.64 | 33.36 | 52.32 | 72.64 | 72.58 |

Table 5: *Global validation accuracy after weight quantization of various quantization robustness variants for Federated Averaging (FEDAVG) on federated version of CIFAR-10 dataset using ResNet-20 architecture.*

| Bit Config | Federated Averaging (FedAvg) | | | | | | |
|---|---|---|---|---|---|---|---|
| | Baseline | KURE | APQN (W-4) | APQN (W-2) | QAT (W-4) | QAT (W-2) | MQAT-W |
| W-32 | 49.17 | 49.57 | 50.77 | 50.8 | 49.19 | 13.62 | 48.47 |
| W-8 | 49.24 | 49.52 | 50.86 | 50.88 | 46.46 | 1.7 | 48.21 |
| W-6 | 48.99 | 49.66 | 50.27 | 50.81 | 47.34 | 1.92 | 48.28 |
| W-4 | 45.37 | 47.75 | 45.76 | 46.34 | 49.7 | 3.05 | 47.18 |
| W-3 | 36.42 | 41.04 | 34.87 | 37.04 | 44.44 | 11.68 | 45.88 |
| W-2 | 6.62 | 8.28 | 8.41 | 7.62 | 24.77 | 42.92 | 42.27 |

Table 6: *Global validation accuracy after weight quantization of various quantization robustness variants for Federated Averaging (FEDAVG) on federated version of CIFAR-100 dataset using ResNet-20 architecture.*

| Bit Config | Federated Averaging (FedAvg) | | | | | | |
|---|---|---|---|---|---|---|---|
| | Baseline | KURE | APQN (W-4) | APQN (W-2) | QAT (W-4) | QAT (W-2) | MQAT-W |
| W-32 | 37.21 | 37.51 | 36.48 | 37.85 | 36.83 | 4.39 | 37.89 |
| W-8 | 37.29 | 37.61 | 36.41 | 37.89 | 36.73 | 2.3 | 37.56 |
| W-6 | 36.61 | 37.17 | 35.9 | 37.35 | 36.98 | 2.73 | 37.23 |
| W-4 | 31.09 | 33.29 | 31.4 | 33.37 | 37.43 | 5.99 | 37.12 |
| W-3 | 17.64 | 17.61 | 20.08 | 19.25 | 31.14 | 15.66 | 36.84 |
| W-2 | 0.86 | 0.9 | 1.02 | 0.89 | 7.28 | 34.53 | 35.45 |

Table 7: *Global validation accuracy after weight quantization of various quantization robustness variants for Federated Averaging (FEDAVG) on federated version of TinyImageNet dataset using ResNet-18 architecture.*

| Bit Config | Federated Averaging (FedAvg) | | | | | | |
|---|---|---|---|---|---|---|---|
| | Baseline | KURE | APQN (W-4) | APQN (W-2) | QAT (W-4) | QAT (W-2) | MQAT-W |
| W-32 | 69 | 69.4 | 69.66 | 68.92 | 65.68 | 27.74 | 65.68 |
| W-8 | 69.02 | 69.4 | 69.66 | 68.86 | 55.1 | 11.52 | 65.66 |
| W-6 | 68.72 | 69.38 | 69.54 | 68.54 | 59.42 | 11.74 | 66.26 |
| W-4 | 68.24 | 68.8 | 68.66 | 68.26 | 66.74 | 12.62 | 66.02 |
| W-3 | 66.82 | 67.12 | 67.06 | 66.94 | 58.4 | 15.36 | 62.51 |
| W-2 | 48.68 | 51.94 | 51.3 | 53.14 | 61.44 | 61.5 | 59.32 |

Table 8: *Global validation accuracy after weight quantization of various quantization robustness variants for Federated Averaging (FEDAVG) on federated version of CIFAR-10 dataset using LeNet-5 architecture.*

| Bit Config | Federated Averaging (FedAvg) | | | | | | |
|---|---|---|---|---|---|---|---|
| | Baseline | KURE | APQN (A-4) | APQN (A-2) | QAT (A-4) | QAT (A-2) | MQAT-A |
| A-32 | 70.16 | 66.82 | 70.42 | 50.56 | 63.74 | 41.04 | 76.28 |
| A-8 | 70.06 | 66.88 | 70.3 | 50.55 | 73.98 | 53.24 | 76.38 |
| A-6 | 70 | 66.78 | 70.1 | 49.98 | 73.72 | 53.22 | 76.42 |
| A-4 | 64.5 | 63.38 | 65.66 | 41.49 | 71.26 | 53.1 | 76.14 |
| A-3 | 47.92 | 50.03 | 51.48 | 22.72 | 51.52 | 53.1 | 74.68 |
| A-2 | 12.66 | 26.2 | 15.2 | 3.8 | 11.06 | 58.2 | 67.78 |

Table 9: *Global validation accuracy after activation quantization of various quantization robustness variants for Federated Averaging (FEDAVG) on federated version of CIFAR-10 dataset using ResNet-20 architecture.*

| Bit Config | Federated Averaging (FedAvg) | | | | | | |
|---|---|---|---|---|---|---|---|
| | Baseline | KURE | APQN (A-4) | APQN (A-2) | QAT (A-4) | QAT (A-2) | MQAT-A |
| A-32 | 49.17 | 37.36 | 50.94 | 50.56 | 18.13 | 4.74 | 40.71 |
| A-8 | 49.37 | 37.22 | 50.86 | 50.55 | 45.62 | 23.91 | 43.79 |
| A-6 | 48.73 | 37.25 | 50 | 49.98 | 45.57 | 23.82 | 44.23 |
| A-4 | 41.36 | 32.5 | 40.6 | 41.4 | 43.06 | 23.13 | 42.59 |
| A-3 | 24.33 | 20.16 | 22.22 | 22.72 | 25.31 | 24.14 | 30.5 |
| A-2 | 4.59 | 3.4 | 4.13 | 3.8 | 1.24 | 7.81 | 28.79 |

Table 10: *Global validation accuracy after activation quantization of various quantization robustness variants for Federated Averaging (FEDAVG) on federated version of CIFAR-100 dataset using ResNet-20 architecture.*

| Bit Config | Federated Averaging (FedAvg) | | | | |
|---|---|---|---|---|---|
| | Baseline | Baseline (Dropout) | QAT (WA-4) | QAT (WA-2) | MQAT-WA |
| WA-32/32 | 70.16 | 72.02 | 60.64 | 26.61 | 74.9 |
| WA-8/8 | 69.86 | 71.62 | 71 | 28.08 | 77.3 |
| WA-6/6 | 69.92 | 70.58 | 71.38 | 29.42 | 76.8 |
| WA-4/4 | 63.12 | 64.08 | 70.04 | 36.44 | 75.54 |
| WA-3/3 | 42.98 | 41.8 | 42.54 | 50.12 | 72.9 |
| WA-2/2 | 11.04 | 14.5 | 9.58 | 40.18 | 66.08 |

Table 11: *Global validation accuracy after activation quantization of various quantization robustness variants for Federated Averaging (FEDAVG) on federated version of CIFAR-10 dataset using ResNet-20 architecture.*

