# OpenReview forum: "Quantization Robust Federated Learning for Efficient Inference on Heterogeneous Devices"
_TMLR — Accepted by TMLR_

### Review · Reviewer_n6mB · 2023-06-16

**Summary Of Contributions:**

The manuscript considers the need for heterogeneous hardware-aware on-device learning and proposes quantization methods for federated learning. The claimed key contributions are
* a multi-bit quantization-aware training (MQAT) over the existing QAT methods (up to the year 2021);
* an additive pseudo-quantization noise
* some theoretical analysis.

**Audience:**

Yes

**Claims And Evidence:**

No

**Requested Changes:**

* The manuscript needs to carefully discuss the related work in the field.
* The manuscript needs to justify the superiority and compatibility of the proposed method.

**Strengths And Weaknesses:**

## Strengths
* The paper is well-structured;
* The studied problem is timely and important.
* The manuscript considers experiments on CIFAR-10/100, TinyImageNet, and FEMNIST.

## Weaknesses
* The manuscript did not discuss the related work well and ignored a significant fraction of prior work on improving federated/distributed learning on heterogeneous hardware systems. See e.g., [1, 2, 3, 4] and their follow-up works (as well as the discussed papers therein).
    * The discussion on quantization-aware training is limited (only consider papers before the year 2021).
    * There is no discussion/comparison of the latest post-training quantization methods, e.g., [5, 6].
* The evaluation did not consider other standard FL methods (w/o considering the hardware heterogeneity), leaving the compatibility of the proposed method unknown.
* The evaluation did not compare with other federated learning methods proposed for heterogeneous hardware.

## Reference
* [1] Heterofl: Computation and communication efficient federated learning for heterogeneous clients, ICLR 2021, https://arxiv.org/abs/2010.01264 (it has more than 200 Google Scholar citations)
* [2] FedLite: A Scalable Approach for Federated Learning on Resource-constrained Clients, http://arxiv.org/abs/2201.11865
* [3] QuPeL: Quantized Personalization with Applications to Federated Learning, http://arxiv.org/abs/2102.11786
* [4] Quantized Distributed Training of Large Models with Convergence Guarantees, http://arxiv.org/abs/2302.02390
* [5] NoisyQuant: Noisy Bias-Enhanced Post-Training Activation Quantization for Vision Transformers, http://arxiv.org/abs/2211.16056
* [6] Optimal Brain Compression: A Framework for Accurate Post-Training Quantization and Pruning, http://arxiv.org/abs/2208.11580

---

> ### Author Response · Authors · 2023-07-16
> **Response to Reviewer n6mB's feedback**
>
> We thank the reviewers for finding our paper well-structured and the topic of the paper timely and important. We thank the reviewer for pointing out more recent relevant works which we will integrate into our manuscript with the following discussion.
>
> ### Related work on FL
> HeteroFL [1] proposes to assign to each client a subset of the global model depending on their resources. A weaker client receives only a subset of hidden layers and as such, it is suited for training as well as inference across a heterogenuous compute landscape. Such an approach is orthogonal to MQAT, similar to how sparsification and quantization are being successfully combined in federated training and centralized settings alike.
>
> QuPeL [3] is very close in scope to our work. A core proposition of their work however is the flexibility of the quantization mechanism to be non-uniform and the set of quantized values that the model weights can occupy to be learnable. Such a method is highly performant in theory but is entirely unsuitable to low-bit accelerators as can be found in today's hardware. Furthermore, QuPeL addresses hardware heterogeneity through personalization, meaning that each client needs to have access a local dataset and perform a finetuning operation in order to select the appropriate centroids for their budget. This is in contrast to our work where the quantization happens ``zero-shot'', i.e., the client does not need any data (i.e., it could be a new client in the federation, only interested in inference) and can just quantize the server model to a specific, hardware friendly, bit-width.
>
> FedLite [2] concerns itself with the compression of activations in the context of federated split learning and as such is out of scope for this work. [4] discusses model and data-parallel training of large language models in a datacenter setting and is thus out of scope of this work. There is no discussion of heterogeneity nor the federated data setting.
>
>
> ### References
>
> [1] Heterofl: Computation and communication efficient federated learning for heterogeneous clients, ICLR 2021.
>
> [2] FedLite: A Scalable Approach for Federated Learning on Resource-constrained Clients, http://arxiv.org/abs/2201.11865
>
> [3] QuPeL: Quantized Personalization with Applications to Federated Learning, http://arxiv.org/abs/2102.11786
>
> [4] Quantized Distributed Training of Large Models with Convergence Guarantees, http://arxiv.org/abs/2302.02390

---

> > ### Author Response · Authors · 2023-07-16
> > **Response to Reviewer n6mB's feedback continued**
> >
> > We have compared against recent and performant post-training quantization schemes for CNNs in our experimental evaluation. Recent works have focused on LLMs and transformers. We will also add the more recent literature for PTQ and QAT in the related work discussion of the final version of the paper with following discussion.
> >
> > ### Recent work on QAT and PTQ
> >
> > Optimal Brain Compression (OBC) [2] extend the Optimal Brain Surgeon (OBS) framework to efficiently quantize and prune NNs in a unified setting. Their method is time- and space-efficient while achieving high accuracy in vision and language models.
> >
> > OPTQ [3] propose an extension of OBC[6] that is optimized for efficient quantization of generative pretrained models. This one-shot weight quantization method that can quantize GPT models with 175 billion parameters in approximately four GPU hours, reducing the bitwidth down to 3 or 4 bits per weight, with negligible accuracy degradation relative to the uncompressed baseline.
> >
> > NoisyQuant [1] address the issue of quantizing heavy-tailed activations in vision transformers. They discover that for a given quantizer adding a fixed uniform noisy bias to the values being quantized can significantly reduce the quantization error. By adding a noisy bias to each layer they are able to actively alter the activations distribution and make it more quantization-friendly.
> >
> > ZeroQuant [6] proposes a method for efficient and accurate PTQ of large-scale transformers. It comprises three main components: (1) a fine-grained hardware-friendly quantization scheme for both weight and activations; (2) a novel affordable layer-by-layer knowledge distillation algorithm (LKD) even without the access to the original training data; (3) a highly-optimized quantization system backend support to remove the quantization/dequantization overhead. ZeroQuant can achieve 8-bit weight/activation quantization of GPT-3-style models with minimal accuracy impact.
> >
> > [5] observe that oscillating latent weights can prevent NNs from converging to optimal solutions during QAT. They propose freezing oscillating weights or dampening oscillations through regularization and thus improve quantized accuracy in efficient ConvNets.
> >
> > NIPQ [4] propose training with pseudo-noise quantization to prevent unstable convergence induced by the straight-through-estimator (STE) in QAT. The NIPQ formulation allows for naturally learning the bitwidth and quantization parameters leading to more accurate and efficient mixed-precision quantized NNs.
> >
> > LLM-QAT [7] investigate QAT for LLMs and propose a data-free distillation method that leverages generations produced by the pre-trained model, which better preserves the original output distribution and allows quantizing any generative model independent of its training data, similar to PTQ methods. They experiment with LLaMA models of sizes 7B, 13B, and 30B, at quantization levels down to 4 bits.
> >
> > ### References
> >
> > [1] NoisyQuant: Noisy Bias-Enhanced Post-Training Activation Quantization for Vision Transformers, http://arxiv.org/abs/2211.16056
> >
> > [2] Optimal Brain Compression: A Framework for Accurate Post-Training Quantization and Pruning, http://arxiv.org/abs/2208.11580
> >
> > [3] OPTQ: Accurate quantization for generative pre-trained transformers. In ICLR, 2023.
> >
> > [4] Nipq: Noise proxy-based integrated pseudo-quantization. In CVPR 2023.
> >
> > [5] Overcoming oscillations in quantization-aware training. In ICML 2022.
> >
> > [6] Zero-quant: Efficient and affordable post-training quantization for large-scale transformers. In NeurIPS 2022.
> >
> > [7] LLM-QAT: Data-Free Quantization Aware Training for Large Language Models. https://arxiv.org/pdf/2305.17888.pdf.

---

### Review · Reviewer_mxjj · 2023-06-21

**Summary Of Contributions:**

This paper considers the problem of Quantization-Aware Training (QAT) in the context of federated learning, to train models which can be easily quantized for on-device inference. This is important since many recent devices require that weights be quantized in order to be able to take advantage of on-device accelerator chips. In the context of federated learning, where the trained model is likely to be deployed on a wide range of devices, this paper specifically considers the problem of training a model that can be robustly quantized to a variety of different bit-widths and quantization step-sizes.

To this end, the paper considers incorporating standard techniques from the model quantization literature (APQN and QAT), while also proposing a straightforward variant of QAT termed MQAT (M for multiple), where at each training iteration a different quantization setting is sampled and used to determine the update. Convergence theory is provided for these three flavors of FedAvg, and experiments illustrate the promise of the proposed MQAT approach on multiple workloads.


**Audience:**

Yes

**Broader Impact Concerns:**

None noted

**Claims And Evidence:**

Yes

**Requested Changes:**

In the revision I would appreciate if the authors can address the following three points:

1. Can the QAT techniques studied in this paper be combined with gradient/update quantization typically used in FL to reduce communication overhead?

2. In Figure 1, what is the intuition for why FedAvg-QAT (W-4) performs well at bit-widths higher than 4, even though it was only trained for 4?

3. In Theorem 1, the term in the upper bound related to QAT depends on the parameter $R$, and for MQAT this depends specifically on the worst-case (largest) quantization step-size, so in theory the performance is no better than QAT for that particular setting. However, the MQAT algorithm uses a randomly sampled bit-width at each iteration. Might one hope to improve the upper bound for MQAT so that it depends on the average quantization step-size (or more generally, something better than the worst-case)?


**Strengths And Weaknesses:**

## Strengths
1. The paper addresses a timely topic and is generally well-written.
2. The paper provides convergence theory, providing a principled foundation for the proposed approaches.
3. The experiments illustrate the benefits of the proposed approaches, and especially MQAT, across several workloads and ablating several aspects of the formulation.

## Weaknesses
1. It is not clear what (if anything) is particularly complicated or a major contribution in the convergence theory, beyond combining convergence techniques for FedAvg with those typically used to analyze QAT and APQN in the non-federated setting. The analysis for MQAT is effectively the same as that of QAT.
2. Some aspects of the experimental results could be investigated and discussed in a bit more depth, to more clearly support the claims  (see below).

---

> ### Author Response · Authors · 2023-07-16
> **Response to Reviewer mxjj's Feedback**
>
> We thank the reviewer for finding the topic of the paper to be timely and we appreciate the reviewer find our paper providing a principled foundation for the proposed approaches. We also appreciate that the reviewer find the experiments and theoretical analysis in the paper to be clearly illustrating the benefit of the proposed approaches. Below we address the reviewer's concerns and suggestions.
>
> ### Training vs. Inference
>
> In this work, we focused on the problem of model quantization for efficient inference with multiple bit-widths. In quantization aware training, the training time is also reduced since both weights and activations are quantized during the forward pass. The quantized training where the gradients are also quantized could be possible and can further improve the training efficiency of training of federated learning. The quantized training is active area of research even in centralized setup and is out of scope of this work. Furthermore, our MQAT variant basically changes the training on client side and thus can be compatible with various versions of FedAvg even where quantized updates of gradients are used to reduce the communication overhead.
>
> ### Upperbound of FedAvg-MQAT
>
> Indeed, thank you for pointing this out. The fact that the bit-width is random can be used to improve the convergence rate for MQAT. More specifically, in Lemma 5 we have that
>
> $\mathbb{E}_b[\mathbb{E}[||\mathbf{r}||^2]] = \sum_d \mathbb{E}_b[\mathbb{E}[r^2_d]] \leq \sum_d \mathbb{E}_b\left[\frac{\Delta_b^2}{4}\right] = \frac{D}{4}\mathbb{E}_b\left[\Delta_b^2\right]
> $
>
> so, depending on the sampling probabilities one has over the bit-widths, the convergence rate can be better than QAT. We will update the submission accordingly.
>
> ### Effectiveness of FedAvg-QAT (W-4) at higher bit-width
> The strength of quantization noise due to the quantization operation in QAT W-4 is not too strong on the models experimented which probably still enables effectiveness at higher bit-width but when the network is trained at QAT W-2, it becomes ineffective at higher bit-width whereas MQAT performs well at various bit-widths.

---

### Review · Reviewer_4rVv · 2023-07-02

**Summary Of Contributions:**

This paper studies model quantization  (quantization-aware training and several variants) in federated learning. The goal is to output a global model that has good average inference accuracy after quantization on each client. It presents multiple algorithms by adapting existing quantization techniques to federated settings, including leveraging Kurtosis Regularization, adding random noise to the weights, and optimizing multi-bit quantization.

The submission presents convergence guarantees of the quantization algorithms, and empirically compares the proposed methods with the baseline of directly quantizing FedAvg models, under different quantization levels.

**Audience:**

Yes

**Broader Impact Concerns:**

None.

**Claims And Evidence:**

Yes

**Requested Changes:**

Fixing 1. and 2. in the 'weaknesses' part.

**Strengths And Weaknesses:**

Strengths:

1. It considers multiple useful techniques to improve quantization-aware training in federated settings. They are not new, but to the best of my knowledge, they are first formally studied in FL in this submission.
2. It gives convergence guarantees for almost all the algorithms.
3. The color coding of the algorithms makes the paper easier to read.

Weaknesses:

1. Writing and motivation need to be improved. The notion of 'robust quantization' is first introduced in the third paragraph of the Introduction, but more context is needed to explain this specific technique or objective. In the introduction, the submission mentions that robust quantization is motivated by randomized smoothing, which would make readers think that the goal of robust quantization is to be robust to adversarial test samples, which is not what this paper means exactly (despite being related). The introduction is also hard to sparse---there are too many technical terms without proper explanation. For instance, what do Kurtosis Regularization, Quantization-Aware Training, Straight Through Estimator offer at a high level. The flow is a bit unnatural. The second paragraph is talking about the need of multi-bit training. But the third paragraph jumps into previous works on robust quantization without mentioning why robust quantization is important, and how it is connected with the previous paragraph.

2. The convergence theorem doesn't cover the convergence of Kurtosis Regularization based objective. It would be interesting to also show Algorithm 1 can converge to the original FedAvg objective. The existing analysis would go through naturally (if the local quantization gradients are unbiased). It would be useful to highlight what is new in the current analysis. Assumption 3 should be written in a more formal form, in consistent with Assumptions 1-2. How does the convergence (along with the error term) scale with the number of local iterations K? How about convex cases? These questions are useful to discuss/answer as well.

---

> ### Author Response · Authors · 2023-07-16
> **Response to Reviewer 4rVv's feedback**
>
> We thank the reviewer for the constructive feedback and finding our paper easy to read and quantization robustness in context of FL to be novel. Below we address reviewer's suggestions.
>
> ### Introduction and Assumption 3
>
> Thanks for the suggestions in relation to Introduction and we will incorporate the suggested changes related to Introduction and Assumption 3 in the final version of the paper.
>
> ### Convergence Analysis
>
> Indeed the convergence theorem doesn't cover KURE, primarily because KURE uses standard unbiased gradients of a regularized objective. This is in contrast to the other methods considered in the paper, which modify the forward pass of the network with either noise injection or quantization and do not add any additional regularizers. In this way, the standard convergence guarantees of FedAvg should apply for the KURE objective, whereas for the other methods one needs to investigate how they affect the convergence of the original, unregularized, objective.
>
> Based on this, the main purpose of our analysis was to show how the gradient bias (relative to the original local gradient where no noise is added and no quantization is performed) due to the changes to the forward pass affects convergence. From Theorem 1 we can see that there is an irreducible noise floor with these methods and the only way to recover the original FedAvg convergence is to increase the bit-width.
>
> Both the convergence term, i.e., the $F(w_1) - F(w^*)$ term, and the the error term scale with the number of local iterations $K$, the first decays with $K$ (which is similar to traditional FedAvg) but some of the error terms increase with $K$, due to each step contributing additional quantization noise and drift between the local and the server weights. We will update the paper with some of this discussion. We targeted specifically the non-convex case, as that aligns with the experiments we performed in the paper.

---

### Review · Reviewer_8f8P · 2023-07-05

**Summary Of Contributions:**

This paper introduces variants of federated averaging with the underlying neural network quantized to various bit-widths with only limited reduction in full precision model accuracy so that models deployed on different devices continue to work despite different computational constraints (i.e., bit-width handling requirements) on those devices. The authors conduct experiments on several benchmarks and provide convergence analysis for their approach, and show promising empirical and theoretical results.

**Audience:**

Yes

**Claims And Evidence:**

No

**Requested Changes:**

see weakness points above regarding framing of the method, strong baselines, and analysis of computational or storage complexity.

**Strengths And Weaknesses:**

Strengths:
1. Well-motivated problem, solution makes sense and is well described. To my knowledge this paper is probably the first to study this problem in-depth through experiments and theoretical analysis.
2. Experiments are conducted on quite a few FL benchmarks
3. Theoretical analysis where appropriate.

Weaknesses:
1. I am not the most familiar with this specific problem, but I was curious as to what is a strong baseline for this problem. Since it is a relatively niche problem, what are the baselines people expect to work? Is the proposed approach a 'strong baseline' (ie taking QAT and extending it to MQAT, and applying to FL), or is it something that we would not have expected to work, and it is cool that it did? If it is the former, phrase it as such and explain why it works with theory and experiments. If it is the second, I would have liked to see what the 'strong baselines' are, and how well they work. Since it's a new problem, I almost don't know how to contextualize how good these results are.
2. There should be some analysis of additional computational or storage complexity given these modifications to the FL methods.

---

> ### Author Response · Authors · 2023-07-16
> **Response to Reviewer 8f8p's feedback**
>
> We thank the reviewer for finding the problem discussed in the paper well-motivated, well-described and novel. We appreciate that reviewer finds our experiments to be exhaustive along with appropriate theoretical analysis to justify the solution. Below we address reviewer's suggestions.
>
> ### Baselines for Robust Quantization in FL
> In this work, we study the quantization of models learnt in FL for efficient inference. This work presents the first of its kind study on efficient models (through quantization) learnt in a decentralized manner. As FL models have heterogeneous hardware requirements, the aim is to learn models robust to quantization at different bit-widths. For this purpose, we utilized existing techniques in centralized training such as KURE [1], APQN [2] and QAT to create strong baselines and introduced MQAT as a solution to training robust models for FL. We would also like to point out that, as far as we are aware, a quantization robust QAT approach such as MQAT has not been introduced prior to this work.  The performance achieved is close to full precision performance which is arguably the performance upper bound one might expect after quantization.
>
> ### Computational or Storage Overhead with robust quantization in FL
> We provide training curves in Fig. 4a of the supplementary and also the theoretical analysis to study the convergence of quantization robust FL trained models. Due to the quantization of weight/activations in the forward pass, locally quantized training is expected to be less computationally expensive compared to the full precision counterpart. Our methods do not have any significant computational or storage overhead since the only change is the quantization operation (Eq. 3) is embedded in forward pass for weights and/or activations. The inference after quantization can be achieved at specific bit-width by using only the quantized tensors to improve the inference latency. This is similar to computational and storage overhead what one expects in the centralized training setup.
>
>
>
> ### References:
>
> [1] Moran Shkolnik, Brian Chmiel, Ron Banner, Gil Shomron, Yury Nahshan, Alex Bronstein, and Uri Weiser. Robust quantization: One model to rule them all. NeurIPS, 2020.
>
> [2] Alexandre Défossez, Yossi Adi, and Gabriel Synnaeve. Differentiable model compression via pseudo quantization noise. arXiv preprint arXiv:2104.09987, 2021.

---

### Decision · Action_Editors · 2023-08-19

**Recommendation:** Accept with minor revision

**Comment:**

The paper aims to improve robustness of FL models to quantization. It presents multiple algorithms by adapting existing quantization techniques to federated settings along with convergence guarantees of the quantization algorithms. Empirical evaluation of the proposed methods leaves more to be desired. We thank the authors and reviewers to actively engage in discussion for making the paper better. During discussion many of the reviewer concerns/questions were resolved like the one about convergence and storage requirements. However some concerns remain about baselines and related works. Nevertheless, the proposed method and technique are correct, experimental result demonstrate robustness to quantization in one way, and the approach will be of interest to the community, hence I propose to accept the paper with some minor modifications.

1. Add evaluation comparing to standard FL methods (w/o considering the hardware heterogeneity), which will allow us to understand the compatibility of the proposed method.
2. Add evaluation comparing with other federated learning methods proposed for heterogeneous hardware.

Finally not required but it would be great if the manuscript could also evaluate the proposed method on the transformer architectures and compare results with that of the corresponding quantization methods.

**Audience:**

Yes, both federated optimization/learning and on-device ML community will be interested in this paper.

**Claims And Evidence:**

Yes

---

> ### Author Response · Authors · 2023-09-18
> **Thanks for the valuable feedback**
>
> Thanks for managing the review of our paper and providing constructing feedback on our paper. We have now added recommended experimental evaluations in the camera ready version of the paper, which are summarized as below:
>
> 1. In order to showcase the compatibility of the proposed quantization robustness mechanism, we performed quantization robustness evaluations at different bit-widths using another standard FL baseline namely, FedProx [a]. Our method has similar improvements as in the case of FedAvg even on FedProx.
> 2. We also performed experimental comparisons against HeteroFL [b] by using BOPs [c] where pruning is used instead of quantization to achieve efficient inference. We report the results in Section 5.1 and we notice that pruning as a mechanism for efficient inference underperforms in comparison to quantization in FL.
> 3. Furthermore, we report quantization robustness evaluation on a transformer architecture i.e., CCT [d] in Section 5.1 to show the applicability of proposed quantization robustness technique on transformer models. Our approach has robustness improvements on CCT as in case of CNNs.
>
> References:
>
> [a] Li et al. Federated Optimization in Heterogeneous Networks. In MLSys 2020.
>
> [b] Diao et al. HeteroFL: Computation and communication efficient federated learning for heterogeneous clients. In ICLR 2021.
>
> [c] Van et al. Bayesian bits: Unifying quantization and pruning. In NeurIPS 2020.
>
> [d] Hassani et al. Escaping the big data paradigm with compact transformers. arXiv 2021.